# Genetic basis of right and left ventricular heart shape

Richard Burns[1], William J. Young [2,3], Nay Aung [2,3,4], Luis R. Lopes [3,5], Perry M. Elliott [3,5], Petros Syrris [5], Roberto Barriales-Villa [6], Catrin Sohrabi[2], Steffen E. Petersen [2,3,4,7], Julia Ramírez [2,8,9,10], Alistair Young [1,10] ✉ & Patricia B. Munroe [2,4,10] ✉

Heart shape captures variation in cardiac structure beyond traditional phenotypes of mass and volume. Although observational studies have demonstrated associations with cardiometabolic risk factors and diseases, its genetic basis is less understood. We utilised cardiovascular magnetic resonance images from 45,683 UK Biobank participants to construct a heart shape atlas from bi-ventricular end-diastolic surface mesh models through principal component (PC) analysis. Genome-wide association studies were performed on the first 11 PCs that captured 83.6% of shape variance. We identified 43 significant loci, 14 were previously unreported for cardiac traits. Genetically predicted PCs were associated with cardiometabolic diseases. In particular two PCs (2 and 3) linked with more spherical ventricles being associated with increased risk of atrial fibrillation. Our study explores the genetic basis of multi-dimensional bi-ventricular heart shape using PCA, reporting new loci and biology, as well as polygenic risk scores for exploring genetic relationships of heart shape with cardiometabolic diseases.

Cardiovascular disease (CVD) is a major burden for the healthcare system, with associated morbidity and mortality[1]. Cardiovascular performance is intrinsically linked to cardiac structure, which is influenced by environmental and disease processes[2], as well as genetic factors[3].

Previously, standard cardiac structural phenotypes have included left ventricular (LV) and right ventricular (RV) mass, volume, mass to volume ratio, sphericity index, conicity and myocardial strain[4–10]. These measures have proven to be important for characterizing CVD and assessing individual outcomes. Higher LV mass to volume ratio for example, is strongly associated with CVD and predictive of clinical events such as heart failure, stroke, coronary heart disease and death[7,11–14]. Left and right end-diastolic volume, end-systolic volume, mass, ejection fraction, concentricity, strain and wall thickness have all demonstrated significant associations with both dilated and hypertrophic cardiomyopathy (DCM and HCM, respectively)[8,9,15]. Sphericity has been associated with atrial fibrillation (AF) and cardiomyopathy[16–19]. Importantly, these cardiac phenotypes have shown high heritability, and several common variants and candidate genes have been identified through genome-wide association studies (GWAS)[3,7,10,20–24]. However, these shape measures are simple one-dimensional metrics and do not capture the multidimensional shape variations that can be extracted from advanced imaging examinations.

[1]School of Biomedical Engineering and Imaging Sciences, King's College London, London, UK. [2] William Harvey Research Institute, Queen Mary University London, Charterhouse Square, London, UK. [3]Barts Heart Centre, St Bartholomew's Hospital, Barts Health NHS Trust, West Smithfield, London, UK. [4] NIHR Barts Biomedical Research Centre, Queen Mary of University London, Charterhouse Square, London, UK. [5]Centre for Heart Muscle Disease, Institute of Cardiovascular Science, University College London, London, UK. [6] Unidad de Cardiopatías Familiares, Cardiology Service, Complexo Hospitalario Universitario A Coruña, INIBIC, A Coruña, Spain. [7] Health Data Research UK, Gibbs Building, 215 Euston Road, London, UK. [8] Aragon Institute of Engineering Research, University of Zaragoza, Zaragoza, Spain. [9]Centro de Investigación Biomédica en Red – Bioingeniería, Biomateriales y Nanomedicina, Zaragoza, Spain. [10]These authors contributed equally: Julia Ramírez, Alistair Young, Patricia B. Munroe. ✉e-mail: Alistair.young@kcl.ac.uk; P.b.munroe@qmul.ac.uk

Statistical shape atlases, constructed from individual heart shapes, provide in-depth quantification of cardiac shape variations[25] and their relationships with cardiovascular risk factors and CVD[26–28] for both the left[29] and right ventricles[30]. Shape variations are more strongly related with cardiovascular risk factors such as hypertension, hyperlipidaemia, diabetes, smoking and obesity[30], and more predictive of major adverse cardiovascular events[2] than standard cardiac structural phenotypes. Genetic associations with heart shape variation may therefore provide information not found from genetic analyses on standard cardiac structural phenotypes and may assist in understanding the mechanisms underlying the development of CVD. Genetic analysis of shape phenotypes has also been previously performed on other biological domains such as facial landmarks[31], illustrating the utility of shape phenotypes in gene discovery.

Here, we constructed a bi-ventricular statistical shape atlas with end-diastolic shape variations from 45,683 UK Biobank participants with cardiac magnetic resonance (CMR) data available. An automated machine learning pipeline was used to detect cardiac landmarks and myocardial contours from CMR images in both short and long axis views. A subdivision surface mesh template was customised to each participant using diffeomorphic non-rigid registration[30,32]. Principal component analysis (PCA) was used to derive principal components (PCs) from the heart shape models to be used as multi-dimensional heart shape phenotypic traits. We estimated the heritability and performed GWAS for these PCs, as well as bioinformatics analyses to identify candidate genes and key biological pathways. Furthermore, we explored associations between the PCs and their polygenic risk scores (PRS) with traditional CMR phenotypes and cardiovascular risk factors and outcomes using correlation metrics and regression analyses.

In this work, we discover there is a significant genetic contribution to PC-derived multidimensional bi-ventricular cardiac shape phenotypes and demonstrate their associations with several prominent cardiometabolic diseases.

## Results

### Bi-ventricular shape analysis pipeline
The CMR images from the 45,683 UK Biobank participants were analysed using an automated image analysis, model customization and statistical atlasing pipeline (Fig. 1). Participants with suitable short and long axis imaging studies had diffeomorphic subdivision mesh models customized at end-diastole, as described previously[30,32]. Of these, 450 imaging studies were incomplete, and 1562 contour sets had missing data; PCA was performed on 43,676 participants mesh models, to identify the independent modes of variation accounting for the greatest amount of shape variation across the cohort. The result is a set of PCs, each with a Z-score value for each participant reflecting their difference from the mean shape in the direction of the PC. These Z-scores between different PCs were uncorrelated.

Since PCA is an unsupervised method of dimensionality reduction, the PCs are not readily interpreted cardiac morphological features. By plotting cardiac shape ± 2 standard deviations from the mean in the direction of each PC, the associated biological shape variation can be visualised (Fig. 2). Animations of these plots are given in the Supplementary Movies 1–33. Descriptively, PC1 was associated with overall heart size, PC2 with apex-base length, PC3 with anterior-posterior width, PC4 with relative orientation of the RV respective to the LV, and PC5 with lateral width. Other PCs had more complicated shape changes; for example, PC9 was associated with relative size and length of the LV respective to the RV. PCs 2, 3 and 5 also represent variations of cardiac sphericity in different dimensions (variation in length in different dimensions causing the ventricles to be more spherical). Together, the first 11 PCs (the PCs that individually captured >1% variance) accounted for 83.6% of the total shape variation; 83.7% in European individuals ($n$ = 41,235), 83.0% in non-Europeans

($n$ = 2,441) and 80.7% in participants with previous myocardial infarction ($n$ = 671).

From the 39,538 individuals with models that passed QC in the atlas (1081 did not pass our QC constraints Fig. 1, Supplementary Methods, 3057 were identified to have erroneous CMR images during modelling), we included 35,055 participants of European ancestry with CMR imaging data and no prior history of cardiovascular disease in subsequent analyses (Supplementary Data 1).

### Correlations between shape PCs and cardiac structural traits, risk factors and disease
To explore the biological manifestations of the PCs, we examined their associations with traditional cardiac structural measures and CVD risk factors using correlation analyses. PC1 was strongly positively correlated with LV and RV structural measures (volumes, mass, stroke volume, $r^2$ 0.81 – 0.94), and negatively correlated with ejection fraction of both ventricles ($r^2$ − 0.16 (left) and −0.22 (right)) and RV mass to volume ratio ($r^2$ − 0.41). Regarding CVD risk factors, PC1 was positively correlated with different anthropometric measures, including height ($r^2$ 0.73), heart rate ($r^2$ 0.32) and blood pressure ($r^2$ 0.16 diastolic, 0.12 systolic), and negatively correlated with age ($r^2$ − 0.13) (Fig. 3, Supplementary Data 2). The other PCs also had significant correlations with these measures.

To ascertain relationships between the PCs and disease, we tested their associations with seven major cardiometabolic diseases: heart failure (HF), myocardial infarction (MI), f (AF, HCM, DCM, 2nd or 3rd degree atrioventricular (AV) block and diabetes, adjusting for covariates using logistic regression (Supplementary Data 3−9 (A)). We found significant odds ratios ($p$ < 0.0071, Bonferroni corrected) per standard deviation increase in PC score with several cardiometabolic diseases, with top associations for AF with PC3 (OR 1.52, $p$ = 2.32 × 10⁻⁶⁰) and PC5 (OR 0.70, $p$ = 2.94 × 10⁻³³), and diabetes for PC1 (OR 0.50, $p$ = 6.17 × 10⁻⁴¹) (Table 1, Supplementary Data 3−9 (A), Fig. 4). Other associations were also observed for HF, DCM, HCM, MI and AV block, in particular PC5 had significant associations with all tested outcomes.

### Heritability and genotypic correlations
All 11 PCs were heritable [$h^2_g$ single nucleotide variant (SNV)], as defined using a variance component analysis with BOLT-REML[33] (Methods), with estimates ranging from 8.5% (PC6) to 36.3% (PC1) (Table 2). The highest genetic correlation amongst PCs was between PCs 6 and 7 ($r_g$ = 0.75) and the lowest between PCs 5 and 11 ($r_g$ = −0.39), summarised in Supplementary Fig. 1.

### Genomic loci associated with biventricular shape
We identified 43 genome-wide significant loci ($p$ < 5 × 10⁻⁸) across 8 PCs with BOLT-LMM[29,34]: 9 for PC1, 4 for PC2, 6 for PC3, 8 for PC4, 9 for PC5, 2 for PC8, 4 for PC9 and 1 for PC10 (Table 3, Supplementary Fig. 3). A locus was defined as a 1Mb region (± 500kb either side) from the lead variant. There was no evidence of confounding from population stratification or cryptic relatedness in our GWAS, as demonstrated by small linkage disequilibrium (LD) score regression intercepts (1.00 − 1.02), low genomic inflation factors ($\lambda$ = 1.03 − 1.12) and quantile-quantile plots (Supplementary Fig. 2). Conditional analyses with genome-wide complex trait analysis (GCTA)[35] (Methods) indicated two additional independent signals, one for PC1 at the BAG3 locus and one for PC3 at PRDM6. In total, 45 independent signals for cardiac shape were discovered at 43 loci, these are summarised in Table 3.

Subsequent analyses focused on the 8 PCs with genome wide significant results: PCs 1−5 and 8−10. For each of these 8 PCs, we performed a lookup of the lead signals (lead SNVs and high LD proxies $r^2$ > 0.8) in the GWAS summary statistics of the other PCs (Supplementary Data 10). Notably, several signals for PC5 (lateral width) were also genome-wide significant in the GWAS for PCs 2 (apex-base length),

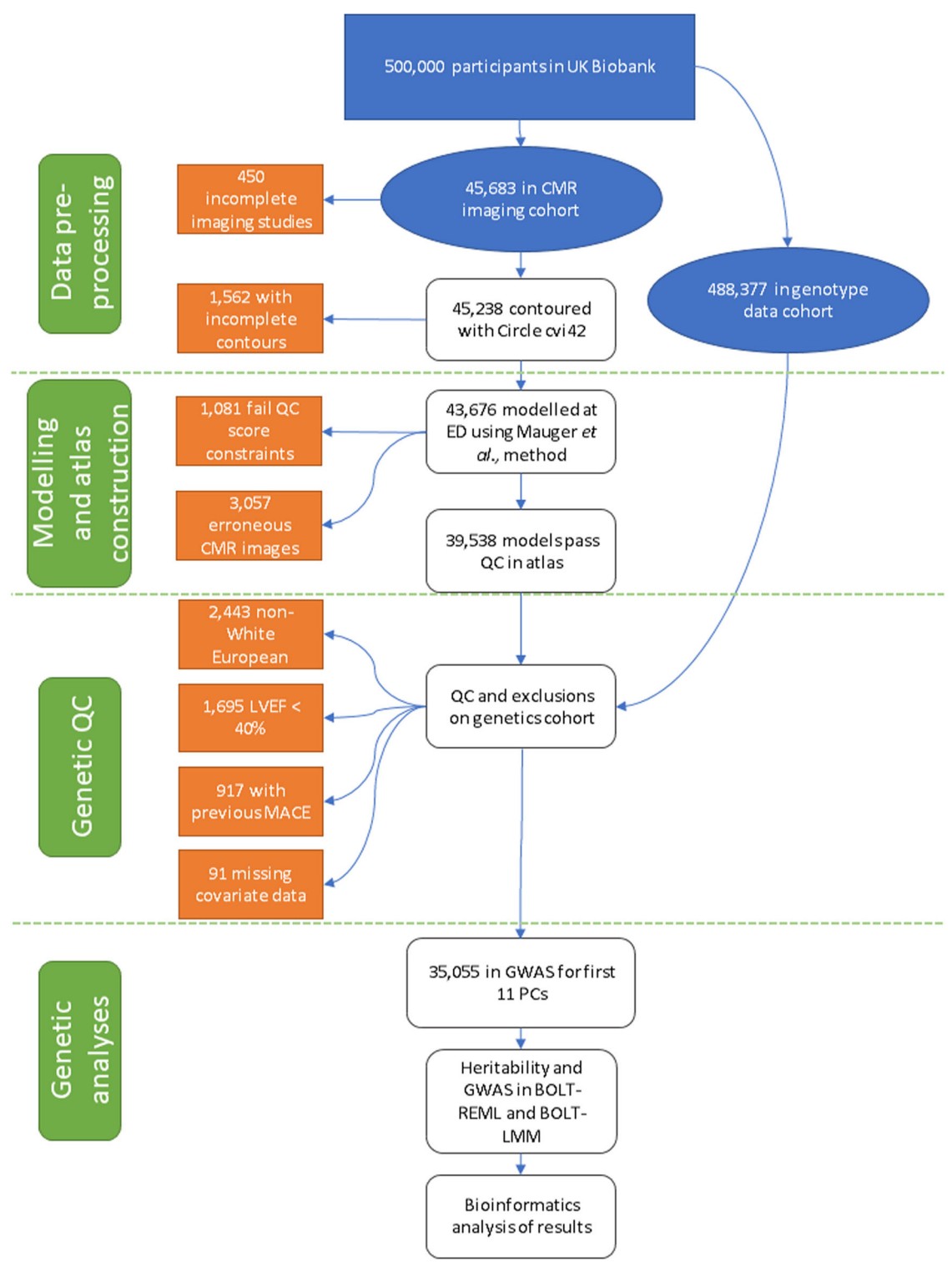

**Fig. 1 | Study design overview.** A flowchart summarizing the study design. Blue highlight indicates study data, green indicates a step in the study, orange indicates an exclusion of data and white describes the methodology applied at this step. QC, quality control; CMR, Cardiovascular magnetic resonance; GWAS, genome-wide association study; LVEF, left-ventricular ejection fraction; MACE, major adverse cardiac event; PC, principal component; ED, end-diastole; atlas creation methodology first described in Mauger et al. method (PMID 30440467). In the Genetic QC section exclusions (orange) are not sequential, and a participant can be in multiple exclusion groups.

3 (anterior-posterior width) and 9 (LV size/length). Across the 45 signals, only 6 were not associated with any other PC ($p \geq 0.05$), four signals for PC1 and one for PC5 and PC9. We also reviewed sharing of loci between PCs, and noted there was no overlap of loci ($p < 0.05$) between six PC pairs. These were PC2 (apex-base length) and PC4 (relative RV orientation), PC4 and PC10 (LV apical displacement in the lateral direction), PC8 (LV morphology) and PCs 2 and 9 (LV size and length) and PC9 with PC1 (heart size).

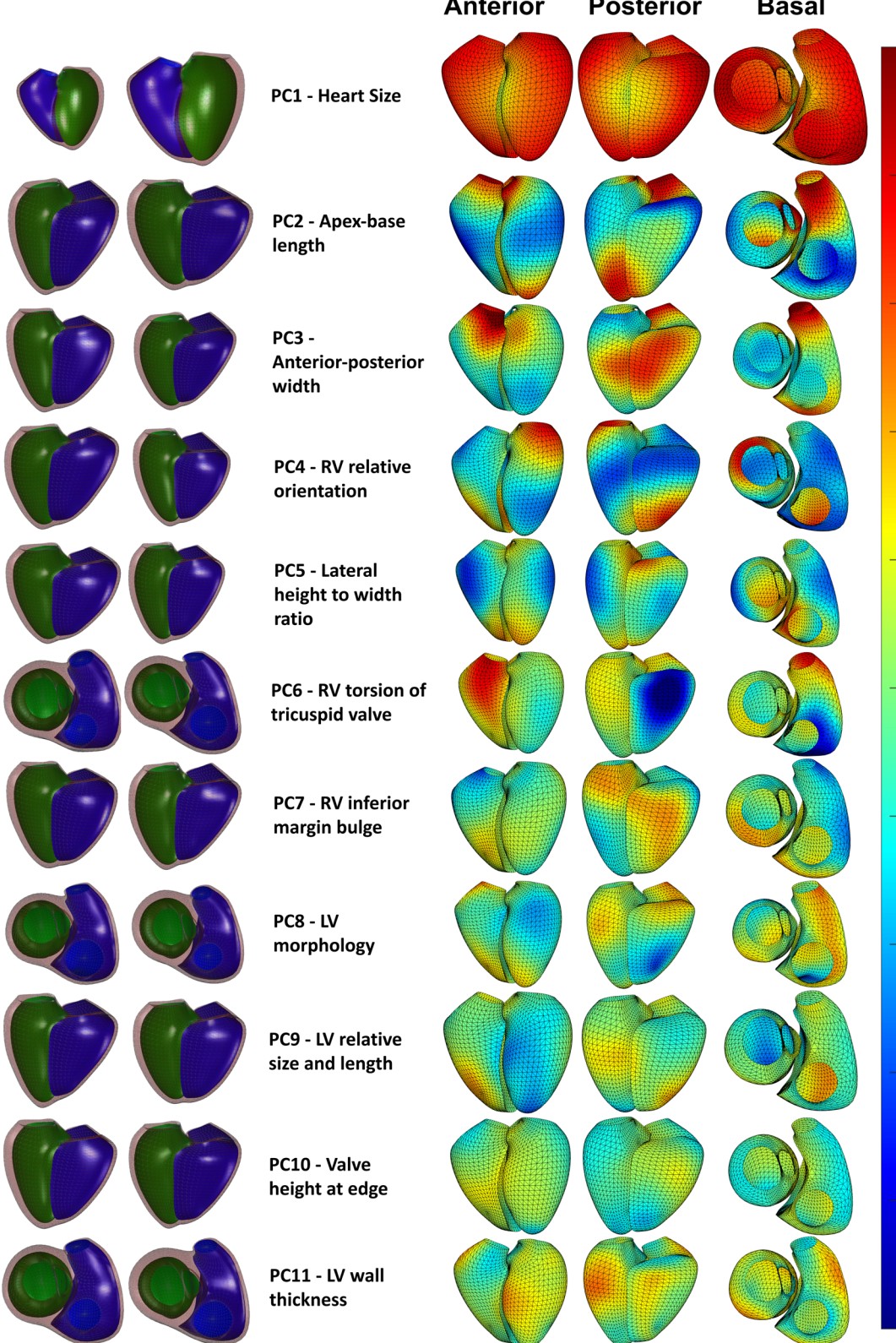

**Fig. 2 | Heart shape component summary.** Visualizations of the heart shape components captured by principal component analysis. From left to right on each row, the following are indicated: −2 and +2 standard deviations from the mean shape of the PC (bottom row), the PC number and estimated biological feature captured, heat-mapped visualizations of the regional shape changes observed in the PC from anterior, posterior, and basal view angles. In this, a warm color indicates topological variation in the positive direction of the PC, and a cool color indicates variation in the negative direction. In this figure we display PCs 1-11. PC, principal component; RV, right ventricular; LV, left ventricular. The figure was created using Matlab R2022a.

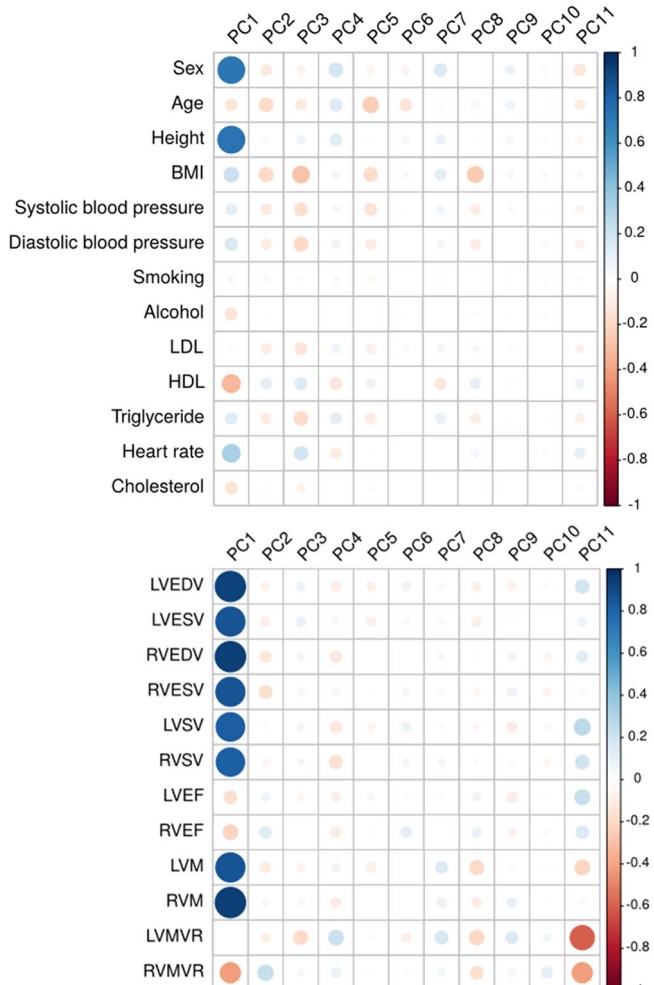

**Fig. 3 | Correlation of PCs with structural measures and cardiovascular disease risk factors.** Plots summarising the Pearson's correlation coefficient. Top is the correlation matrix between shape PCs and cardiovascular disease risk factors, below is the correlation matrix between shape PCs and CMR-derived measures of cardiac structure and function. ($n = 39,538$) *In each correlation matrix the size of the circle indicates the p-value significance (the smallest circle is p < 0.05), and the colour of the circle indicates the strength of the Pearson's correlation coefficient.* The correlation test is two-sided in this analysis. PC, principal component; CMR, cardiovascular magnetic resonance; LV, left ventricular; RV, right ventricular; EDV, end-diastolic volume; ESV, end-systolic volume; SV, stroke volume; EF, ejection fraction; (LV/RV), ventricular mass; MVR, mass to volume ratio; BMI, body mass index; LDL, low density lipoprotein levels; HDL, high density lipoprotein levels; Triglyceride, blood triglyceride levels.

We also performed a SNV-wise gene analysis using multi-marker analysis of genomic annotation, (MAGMA; *Methods*), which combines SNV p-values at a gene into a gene test-statistic with LD adjustments[36]. This analysis identified 85 genes (Supplementary Data 11) across the 8 PCs ($p < 2.6 \times 10^{-6}$, Bonferroni corrected).

## Variant functional annotation
The majority of variants were non-coding, however there were four non-synonymous missense variants; one was predicted to be "possibly damaging" (rs2279472, chr2:179672414:C > T, *TTN*) with Variant Effect Predictor[37–39] (*Methods*, Supplementary Data 12).

We identified 35 variants as expression quantitative trait loci (eQTLs) in LV, aortic artery, coronary artery and atrial appendage tissues using the Genotype-Tissue Expression project (GTEx, *Methods*)[40]. Twenty-eight had support for colocalization (PP4 > 0.75, *Methods*)[41] at

nine of our lead SNVs (Supplementary Data 13). For both PCs 1 and 9 there were two loci with colocalised eQTLs. For PC1, there was an eQTL associated with expression of *VEGFB* in LV and atrial appendage tissues, and an eQTL for 4 genes (*BHMG1* for atrial appendage tissue, *SYMPK* for LV and atrial appendage tissues and *SIX5* for aortic artery tissue). For PC9, there was an eQTL for *CEP85L* in aortic artery tissue, and one for *FBN2* in LV tissue.

To assess the effects of changes in gene expression in LV, atrial appendage, coronary artery and aortic artery tissues on the PCs, we employed transcriptome-wide analysis of predicted gene expression with S-PREDIXCAN[42] (*Methods*). We identified 19 genes ($p < 3.1 \times 10^{-6}$, Bonferroni corrected) mapping to the boundaries of 11 loci, of which 3 were significant in multiple cardiovascular tissues (Supplementary Data 14).

## Long-range tissue-specific interactions of shape PC variants
Potential target genes of regulatory SNVs were identified with long-range chromatin interaction analyses (Hi-C) in LV, RV and aortic tissues (*Methods*). We identified 50 significant promoter interactions (RegulomeDB score ≤ 2)[43] (Supplementary Data 15 and 16). Twenty-six unique loci had long range interactions in LV and RV tissues; for PC4, we observed 15 interactions at 5 loci. At the *WNT16* locus there were long range interactions in LV tissue with *ING3*, *ANKRD7* and *TSPAN12*, and in both LV and RV tissues with *CPED1*. At the *CCDC91* locus, there were long-range interactions in LV tissue with *TSPAN11*, *IPO8*, *PTHLH*, *CAPRIN2* and *CCDC91* itself.

We also tested for enrichment of variants in DNase I hypersensitivity sites using the GWAS Analysis of Regulatory and Functional Information Enrichment with LD correction tool (GARFIELD)[44]. We did not observe any significant enrichment when examining genome-wide significant SNVs, however significant enrichment was observed in fetal heart tissue for PC2 and PC5, when including suggestive SNVs ($p < 1 \times 10^{-6}$) (Supplementary Data 17, Supplementary Fig. 4).

## Candidate gene identification and pathway analysis
Candidate genes at each locus were prioritised using an integrative approach that utilised multiple lines of evidence (*Methods*): proximity to lead variant, presence of missense variants, evidence from eQTL and Hi-C analyses, support from mouse knockout models from the mouse genome informatics database[45], literature review for cardiovascular disease association and genes identified from MAGMA output. Support for selection of a candidate gene at a locus was based on a minimum of 2 lines of evidence. Using these criteria, we identified a total of 69 candidate genes for the 45 signals (Supplementary Data 18).

We identified enriched pathways using the FUMA Gene2Func tool[46]. Using genes for each PC individually in pathway analysis resulted in nominally significant results for some PCs (Supplementary Data 19), therefore we performed a pathway analysis including all 69 candidate genes from across all PCs. A total of 312 gene ontology biological processes were significant (False discovery rate (FDR) $p < 0.05$, Supplementary Data 20). The top results were cardiac related terms such as heart processes ($p = 2.14 \times 10^{-12}$), circulatory system development ($p = 4.27 \times 10^{-12}$) and striated muscle contraction ($p = 1.56 \times 10^{-11}$). There were also significant results for cellular components associated with muscular contraction.

## Pleiotropy with cardiovascular traits
We investigated associations between our 45 lead and conditionally independent variants and those in high LD ($r^2 > 0.8$) with cardiac-associated traits in the GWAS Catalog[47], Phenoscanner[48,49] and across published cardiovascular GWAS not present in these databases to ascertain pleiotropy and potential novelty of cardiac shape loci. We identified several associations with established cardiac structural, functional and disease traits (Supplementary Data 21 and 22). For example, a variant at the *BAG3* locus (rs72840788), which is a

**Table 1 | Overview of shape PCs and associations with cardiometabolic outcomes**

| PC | (%) Shape variance captured | Estimated biological feature of the PC | Cardiometabolic disease associations |
|---|---|---|---|
| 1 | 34.4 | Overall heart size | Diabetes, HF, AV block, DCM, AF, MI |
| 2 | 12.4 | Apex to base length | AF |
| 3 | 9.7 | Anterior-posterior width | AF, HF, HCM, DCM, MI, Diabetes |
| 4 | 8.7 | Relative right ventricular orientation | AF, HF, MI, Diabetes |
| 5 | 6.3 | Lateral height to width ratio | AF, HF, DCM, MI, HCM, AV block, Diabetes |
| 6 | 4.5 | Right ventricular torsion of the tricuspid valve | AF |
| 7 | 2.4 | Right ventricular inferior margin bulging | AF, Diabetes |
| 8 | 1.6 | Left ventricular morphology | MI, HF, AV block, DCM, Diabetes |
| 9 | 1.3 | Left ventricular size and length | HCM, AV block, HF, MI, DCM |
| 10 | 1.2 | Left ventricular apical displacement in the lateral direction | Diabetes, HF, AF |
| 11 | 1.2 | Left ventricular free wall thickness | AF, Diabetes |

PC, principal component; HF, heart failure; DCM, dilated cardiomyopathy; HCM, hypertrophic cardiomyopathy; AF, atrial fibrillation; MI, myocardial infarction; AV block, atrio-ventricular block (2nd or 3rd degree). All associations were $p < 0.0071$ Bonferroni corrected.

secondary signal in PC1, has been previously identified for heart function traits (LV ejection fraction, LV systolic and diastolic function) and HCM. For PC1, shared genetic variants with diastolic blood pressure (*ATXN2, GOSR2, NOS3, BAG3*) and hypertension (*ATXN2, VEGFB, NOS3, BAG3*) were also observed.

To identify signals that were not previously reported for CV (cardiovascular) traits, for each PC we performed a lookup of the lead SNV at the signal and all variants within the locus ±500kb with $r^2 > 0.1$ for association with a GWAS SNV for a CV trait. Fourteen of the 45 signals had not been previously reported in a GWAS of a CV trait. The 14 signals are attributed to the following candidate genes: *FAAP24* for PC2, *CHTOP, FERD3L* and *DDC* for PC3, *CCDC91, WNT16, C7orf25, TBX18, GMDS* and *LINC02873* for PC4, *EN1* and *SIM2* for PC5 and *EPHB2* and *H1-0* for PC8 (Table 1 *and* Supplementary Data 23). Seven of these variants had associations with non-cardiac phenotypes, the remaining 7 signals were not previously identified in any GWAS (*Methods*).

## PheWAS and polygenic risk scoring
To examine relationships across the 8 PCs and clinical conditions agnostically, we performed a phenome-wide association study (PheWAS) in the UK Biobank using our lead and secondary variants[50]. From 325 phenotypes with at least 200 cases available we observed two significant ($p = 1.54 \times 10^{-4}$, Bonferroni corrected) associations (*Methods*). The variant rs791274 for PC5 was associated with rheumatic disease of the heart valves, and rs7620927 for PC4 with respiratory symptoms (Supplementary Data 24*)*.

We next assessed relationships of genetically predicted PCs with the cardiometabolic diseases previously tested with each phenotype. This analysis was performed in unrelated UK Biobank individuals of European ancestry that were not included in the GWAS ($n = 371,264$). Polygenic risk scores (PRS) for shape PCs 1–5 and 9 were calculated using the software PRSice2 (version 2.3.5)[51]. There were limited loci for PCs 8 and 10, and none for PCs 6 and 7 so PRS analysis was not performed for these PCs. First, we confirmed each PRS was positively correlated with their corresponding PC ($r = 0.043 - 0.076$, $p = 1.29 \times 10^{-6} - 2.50 \times 10^{-47}$). We then tested the standardised PC PRS for associations with HF, MI, AF, HCM and DCM, 2nd or 3rd degree AV block and diabetes, with covariate adjustment (*Methods*). We identified several significant associations ($p$-value threshold $< 0.0071$, Bonferroni corrected, Fig. 5, Supplementary Data 3-9 (B)). PC1 (larger heart size) was associated with reduced odds of MI (OR 0.95) and diabetes (OR 0.94), PC2 (longer apex-base length) was associated with reduced odds of HF (OR 0.97) and AF (OR 0.95), PC3 (greater anterior-posterior width) was associated with increased risk of AF (OR 1.04), PC5 (smaller lateral width) was associated with increased risk of HCM (OR 1.29), and reduced risk of diabetes (OR 0.97) and HF (OR 0.98), and PC9 (longer

LV with respect to RV) was associated with reduced risk of AF (OR 0.96), MI (OR 0.98) and AV block (OR 0.95) (Supplementary Data 3–9 (B)).

As there was discordance in the directions of effect between the observational and genetic associations between PC5 and HCM, we sought to replicate the PC5 PRS HCM association. We validated the positive association between the PC5 PRS and HCM in an external European HCM cohort comprising 2,284 cases and age and sex matched UK Biobank controls, with OR of 1.21 (95% confidence intervals 1.16–1.27, $p = 1.63 \times 10^{-19}$).

We subsequently tested the PC PRS in a multi-ancestry cohort from UK Biobank (Total $n = 388,152$, European (not included in GWAS) = 371,264, African = 6716, South Asian = 8501, East Asian = 1671). The same significant associations between the PRS and diseases remained in the multi-ancestry dataset (Supplementary Data 3b–9b). We next investigated the allele frequencies of the SNVs included in the PC PRS across ancestries. For the majority of lead variants there was no significant deviation in allele frequencies across ancestries, however several of the variants for PC1 had very low frequencies in individuals of African and East Asian ancestry ( < 1%) (Supplementary Data 25).

We also examined the percentage of variance observed in the PCs across different ancestries that could be explained by the European PRS (Supplementary Data 26). For PC1 and PC4 the PRS explained more variance in Europeans than in the other ancestries, this can be explained by up to 50% of the variants in the PRS being rare in other ancestries. However, for PCs 2,3, 5 and 9, we observed some PRS explained more variance in samples of non-European ancestry compared to Europeans (e.g., PC9 explained only 0.38% of trait variance in Europeans, but 6.35% in East Asians, Supplementary Data 26). There were only 4 variants in the PRS for PC9, and the allele frequencies (Supplementary Data 25) for Europeans and East Asians varied largely for three of the four variants, which may explain these differences.

## Mendelian randomization
To further examine the relationships observed between the most significant shape-disease associations identified through observational analyses, i.e. between PC1 and T2D, and between PCs 3 and 5 and AF (Supplementary Data 3a–9a), we performed bidirectional two-sample Mendelian randomization (MR) analyses. When considering T2D as the exposure variable and PC1 as the outcome, we found evidence for a negative causal effect using the IVW method (OR of 0.595, 95% CI 0.228-0.962, $P$-value = 0.0074), as well as MR-PRESSO (OR 0.968, 95% CI 0.947-0.990, $P$-value = 0.021) and others (Supplementary Data 27). We did not find evidence for a significant causal effect of PC1, when considered as the exposure, on T2D, when considered as the outcome.

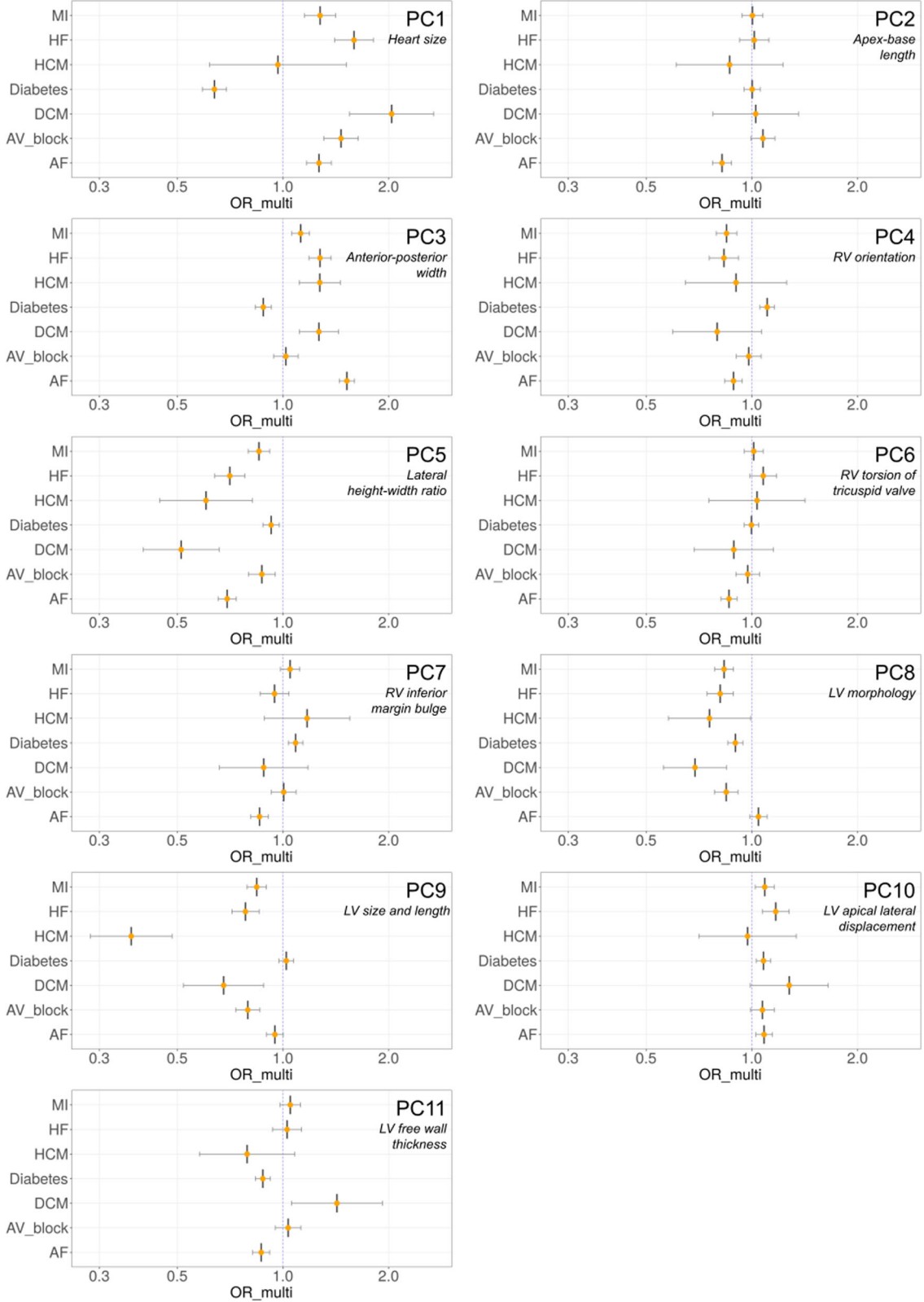

**Fig. 4 | Odds-ratio plots for relationship between PC score and cardiometabolic diseases.** Bar plots illustrating the odds ratios from the logistic regression of PC scores against the incidence of cardiometabolic diseases, after adjusting for relevant covariates (*n* = 39,538). A vertical blue dotted line is plotted at an odds-ratio of 1, with each individual disease having an odds-ratio plotted as a central vertical line and orange point, and confidence interval lines flanking as an intersecting horizontal line denoting 95% confidence intervals. PC, principal component; OR, odds ratio; MI, myocardial infarction; HF, heart failure; HCM, hypertrophic cardiomyopathy; Diabetes, diabetes mellitus; DCM, dilated cardiomyopathy; AV block, atrioventricular block; AF, atrial fibrillation; ICD codes used to define these diseases in Supplementary Data 28.

**Table 2 | Heritability estimates for shape PCs**

| PC | Heritability (%) | Genome-wide significant SNVs ($p < 5 \times 10^{-8}$) | | Genome-wide significant & suggestive SNVs ($p < 1 \times 10^{-6}$) | |
|---|---|---|---|---|---|
| | | Variance explained (%) | Heritability explained (%) | Variance explained (%) | Heritability explained (%) |
| 1 | 36.3 | 1.1 | 3.0 | 2.6 | 7.1 |
| 2 | 24.2 | 0.5 | 2.2 | 1.5 | 6.4 |
| 3 | 26.3 | 0.6 | 2.3 | 2.6 | 10.0 |
| 4 | 32.9 | 1.0 | 2.9 | 3.1 | 10.0 |
| 5 | 27.3 | 1.1 | 3.9 | 2.5 | 9.3 |
| 6 | 8.5 | - | - | 0.5 | 5.7 |
| 7 | 10.2 | - | - | 0.5 | 4.7 |
| 8 | 14.0 | 0.2 | 1.4 | 0.7 | 4.8 |
| 9 | 25.2 | 0.5 | 1.8 | 1.5 | 6.1 |
| 10 | 12.1 | 0.1 | 0.8 | 0.6 | 5.1 |
| 11 | 13.4 | - | - | 0.8 | 6.0 |

Genome-wide significant SNVs indicates all SNVs identified through genome-wide association study with a $p$-value $< 5 \times 10^{-8}$, and suggestive SNVs are all SNVs with $p$-value $< 1 \times 10^{-6}$. PC, principal component; SNV, single nucleotide variant; - indicates no result.

\* For PC11 with no genome wide significant SNVs and small % variance explained by the phenotype in the atlas no work was done looking at the suggestive SNVs.

We also did not find support for a causal association between AF and PCs 3 or 5 (Supplementary Data 27).

## Discussion

This is the first study to examine the genetic architecture of multidimensional biventricular heart shape using PCA. We found the majority of the PCs, reflecting different characteristics of heart shape, to be moderately heritable, consistent with previous traditional measures of cardiac structure. We identified 45 signals across 8 PCs, of which 14 had not previously been reported for any cardiac trait, and 6 which were not associated with any other PC. The PCs demonstrated significant associations with cardiometabolic diseases, including AF and MI, and genetically predicted PCs were generally consistent with these observations.

Our first finding is that the PCs capture additional and distinct information on heart shape compared with previous measures of heart structure and function. Although we observed statistical correlation between previous measures of cardiac structure and our PCs (e.g., PC1, reflecting ventricular volume, and heart size), the strength of this correlation was often weak (Supplementary Data 2), suggesting that the PCs are not simply proxies for existing measures, and supporting the hypothesis that our statistical atlasing method captures distinct heart shape variation. This is further supported by the limited pleiotropy in our GWAS results; of the 45 signals identified in our GWAS, 14 are not previously reported for CV traits, and of these 7 have not been previously reported in any GWAS. The advantages of using the PCs as a shape measure is that they intrinsically capture significant amounts of shape variation across the cohort. Other non-traditional shape measures include trabeculation[52], which captures information independent of the current analysis (since trabeculations are ignored by the image analysis and shape modelling pipeline).

Our GWAS for heart shape has provided several interesting candidate genes at signals for PCs 1 and 5 which are not reported for other CV traits. For PC1, signals were found near *VEGFA* and *VEGFB*, two members of the PDGF/VEGF growth factor family, which induce proliferation and migration of vascular endothelial cells and are essential in physiological and pathological angiogenesis[53]. The signals at these two loci were unique to PC1. At another unique signal for PC1, candidate genes include *NOS3* and *BAG3*, both well characterised proteins

associated with cardiovascular traits and disease, and the nitric oxide produced from *NOS3* is an important mediator of *VEGF*-induced angiogenesis[7,54,55]. At a unique signal for PC5, two candidate genes are indicated, neither gene at the locus (*SIM2* and *CLDN14*) has any prior association with cardiac phenotypes.

Other genes of interest include T-Box Transcription Factor 18 (*TBX18*). We found a novel signal at this locus for PC4 not previously reported for cardiac traits (Supplementary Data 18, 23). *TBX18* is a member of an evolutionarily conserved family of transcription factors that play an essential role in embryonic development. Other variants in the *TBX18* gene have previously been identified in GWAS for HCM and various cardiac disorder related death causes[56]. The protein encoded by this gene acts as a transcriptional repressor involved in the developmental processes of a variety of tissues and organs, including the heart and coronary vessels, and is required for embryonic development of the sinoatrial node head area[57]. Additionally, for PC4 at a locus not previously reported for any cardiac trait is Wingless-Type MMTV Integration Site Family, Member 16 (*WNT16*). The *WNT16* gene belongs to the WNT family of structural-related genes encoding secreted signalling proteins, which have been implicated in developmental processes including embryogenesis[58]. In mouse models, *WNT16* has been identified as a regulator of vascular smooth muscle contraction[59], and in rat models *WNT16* shows reparative and regenerative effects in infarcted hearts[60].

Our results are also consistent with previous findings in smaller cohorts, as we observed many of the PCs to be associated with other well-established CV risk factors in UK Biobank[30]. In prior work, overall larger heart size has been associated with increased adverse outcomes[2,61,62]. Using logistic regression, we found that an increase in PC1 (a larger heart) was associated with an increased risk of HF, AF, DCM, MI and atrioventricular block. Moreover, we observed that an increase in PC1 and the PC1 PRS were significantly associated with a reduced risk of T2D, in line with previous studies reporting diabetes is associated with smaller hearts[63], and through MR it was established that T2D has a significant negative causal relationship with overall heart size (PC1).

We observed that more spherical hearts, and their genetic predisposition, are associated with adverse disease events, with AF being the outcome most significantly associated with sphericity-associated PCs. Cardiac sphericity is a measure of the overall roundness of the heart's chambers, and an established shape-based risk marker for CVDs such as MI[16,17], ventricular arrhythmias[18], cardiomyopathies[17], HF and AF[19] with rounder, more spherical hearts conferring greater risk. Previous studies have only evaluated sphericity as a one-dimensional metric, rather than a high-dimensional shape feature, thus potentially ignoring relevant information. For example, Vukadinovic et al.[10] identified four loci associated with LV sphericity index (*PLN, ANGPT1, PDZRN3,* and *HLA DR/DQ*) in 38,897 UK Biobank participants. Two of these variants were identified in our ED shape GWAS, alongside a further 18 genetic signals associated with sphericity (in PCs 2, 3 and 5). These variants not being identified in previous genetic analyses of sphericity indicates that our multidimensional characterisation of shape offers more power to examine the genetic basis of cardiac shape features such as sphericity than one-dimensional metrics.

Our results were mostly concordant for association between the PCs and their PRSs with cardiometabolic diseases. However, PC5 (the measure of lateral width of the heart) showed an opposite direction of effect with HCM than its PRS, this latter association validated in an external dataset. A possible explanation of this discordance may be the low number of HCM cases ($n = 36$, 0.08% of cohort) in the UK Biobank European imaging cohort ($n = 41,235$), and further work in larger datasets will be required to confirm this. Finally, we demonstrate potential applicability of our multidimensional heart shape PRS with results being consistent across European and multi-ancestry individuals (Supplementary Data 25, 26). Overall, the PRS we have developed

**Table 3 | Genomic loci identified for shape PCs**

| PC | Nearest Gene | Lead variant | CHR | Position (Hg19) | EA | NEA | EAF | BETA | SE | *p* |
|---|---|---|---|---|---|---|---|---|---|---|
| 1 | ATXN2 [c] | rs653178 | 12 | 112007756 | C | T | 0.48 | −0.032 | 0.004 | 4.90E − 17 |
| 1 | TTN [c] | rs2279472 | 2 | 179672414 | C | T | 0.05 | 0.072 | 0.009 | 2.00E − 15 |
| 1 | RSPH6A | rs12460541 | 19 | 46312077 | G | A | 0.35 | −0.027 | 0.004 | 3.80E − 11 |
| 1 | GOSR2 [c] | rs533030436 | 17 | 45091770 | A | G | 0.12 | −0.038 | 0.006 | 9.70E − 10 |
| 1 | VEGFB | rs56271783 | 11 | 64004723 | G | C | 0.05 | 0.053 | 0.009 | 6.50E − 09 |
| 1 | HLA-DRB5 | rs7773935 | 6 | 31228278 | G | A | 0.49 | 0.022 | 0.004 | 1.00E − 08 |
| 1 | VEGFA | rs2146324 | 6 | 43756863 | A | C | 0.26 | 0.025 | 0.004 | 1.30E − 08 |
| 1 | NOS3 | rs3918226 | 7 | 150690176 | C | T | 0.08 | 0.04 | 0.007 | 1.70E − 08 |
| 1 | TIAL1 | 10:121347839_TTTTC_T | 10 | 121347839 | TTTTC | T | 0.23 | 0.025 | 0.005 | 2.80E − 08 |
| 2 | CDC27 [c] | rs117953218 | 17 | 45244074 | T | C | 0.14 | 0.069 | 0.007 | 4.30E − 21 |
| 2 | FAAP24 [a,b] | rs34723366 | 19 | 33463247 | G | T | 0.18 | 0.038 | 0.006 | 6.80E − 09 |
| 2 | CAV1 | rs9886216 | 7 | 116191697 | A | G | 0.25 | −0.033 | 0.006 | 1.10E − 08 |
| 2 | PALLD | rs10155248 | 4 | 169666162 | T | A | 0.47 | −0.028 | 0.005 | 2.40E − 08 |
| 3 | MYH6 [c] | rs452036 | 14 | 23865885 | G | A | 0.35 | −0.043 | 0.007 | 4.80E − 11 |
| 3 | CHTOP [a] | rs749508246 | 1 | 153623678 | TATAGATAG | T | 0.49 | 0.039 | 0.006 | 1.30E − 09 |
| 3 | CDKN1A [c] | rs3176326 | 6 | 36647289 | G | A | 0.2 | 0.045 | 0.008 | 1.1E − 8 |
| 3 | FERD3L [a] | rs10950714 | 7 | 19370645 | A | G | 0.41 | 0.036 | 0.006 | 1.60E − 08 |
| 3 | DDC [a] | rs11238134 | 7 | 50542927 | A | C | 0.33 | 0.037 | 0.007 | 2.30E − 08 |
| 3 | PRDM6 | 5:122551296_TC_T | 5 | 122551296 | TC | T | 0.03 | −0.104 | 0.019 | 3.60E − 08 |
| 4 | CCDC91 [a,b,c] | 12:28407705_CT_C | 12 | 28407705 | CT | C | 0.27 | −0.058 | 0.007 | 6.20E − 16 |
| 4 | WNT16 [a,c] | rs3801387 | 7 | 120974765 | A | G | 0.28 | 0.051 | 0.007 | 4.90E − 13 |
| 4 | RSRC1 [c] | rs7620927 | 3 | 158245883 | A | G | 0.48 | 0.044 | 0.006 | 2.50E − 12 |
| 4 | C7orf25 [a] | 7:42696499_AG_A | 7 | 42696499 | AG | A | 0.28 | −0.043 | 0.007 | 5.80E − 10 |
| 4 | TBX18 [a,b] | rs4466228 | 6 | 85174191 | G | A | 0.16 | 0.05 | 0.009 | 7.30E − 09 |
| 4 | GMDS [a,c] | rs767102318 | 6 | 1904011 | GC | G | 0.33 | −0.04 | 0.007 | 7.60E − 09 |
| 4 | LINC02873 [a,b] | rs4937553 | 11 | 130500876 | G | A | 0.33 | 0.038 | 0.007 | 1.50E − 08 |
| 4 | MIR9-1HG | rs4414033 | 1 | 156406853 | G | A | 0.38 | 0.036 | 0.007 | 3.60E − 08 |
| 5 | SLC35F1 [c] | rs3951016 | 6 | 118559658 | T | A | 0.47 | −0.047 | 0.006 | 1.90E − 15 |
| 5 | STRN [c] | rs10193295 | 2 | 37169049 | G | A | 0.47 | −0.044 | 0.006 | 4.80E − 14 |
| 5 | EN1 [a,b,c] | rs332101 | 2 | 119481738 | A | G | 0.32 | 0.043 | 0.006 | 4.90E − 12 |
| 5 | RAB44 [c] | rs791274 | 1 | 147234095 | C | T | 0.44 | 0.036 | 0.006 | 9.80E − 10 |
| 5 | PALLD | rs10155223 | 4 | 169666161 | A | T | 0.47 | −0.036 | 0.006 | 1.20E − 09 |
| 5 | CDKN1A [c] | rs146170154 | 6 | 36646768 | C | CTA | 0.2 | −0.042 | 0.007 | 9.0E − 9 |
| 5 | ACTN2 [c] | rs4659701 | 1 | 236853167 | G | A | 0.37 | 0.035 | 0.006 | 9.70E − 09 |
| 5 | SETBP1 | rs669738 | 18 | 42465957 | A | C | 0.45 | −0.033 | 0.006 | 2.30E − 08 |
| 5 | SIM2 [a,b] | rs57655466 | 21 | 38021157 | A | C | 0.31 | −0.035 | 0.006 | 4.40E − 08 |
| 8 | EPHB2 [a,b,c] | rs35001652 | 1 | 23082667 | G | A | 0.37 | 0.042 | 0.007 | 1.90E − 09 |
| 8 | H1-0 [a] | rs11703407 | 22 | 38200124 | C | T | 0.3 | 0.042 | 0.007 | 6.60E − 09 |
| 9 | NKX2-5 | rs55676951 | 5 | 172640280 | A | G | 0.25 | −0.057 | 0.008 | 4.90E − 12 |
| 9 | SLC35F1 [c] | rs56403768 | 6 | 118703534 | T | C | 0.42 | −0.049 | 0.007 | 6.70E − 12 |
| 9 | FBN2 | rs4836390 | 5 | 128011688 | T | C | 0.17 | 0.052 | 0.01 | 4.10E − 08 |
| 9 | ANGPT1 [c] | rs1461990 | 8 | 108087628 | C | G | 0.5 | −0.039 | 0.007 | 4.30E − 08 |
| 10 | NOS1AP | rs10918594 | 1 | 162030688 | C | G | 0.33 | 0.043 | 0.008 | 1.50E − 08 |
| Conditionally independent secondary signals | | | | | | | | | | |
| 1 | BAG3 | rs72840788 | 10 | 121415685 | G | A | 0.22 | 0.023 | 0.005 | 7.6E − 7 |
| 3 | PRDM6 | rs10075071 | 5 | 122419344 | C | A | 0.42 | −0.034 | 0.006 | 1.2E − 7 |

*PC, Principal component; CHR, chromosome; Hg19, Genome Reference Consortium Human Build 37 (GRCh37); EA, effect allele; NEA, non-effect allele; Beta, effect size; SE, standard error; p, probability value;*

[a] *denotes signal not previously reported for cardiac traits at time of submission,*

[b] *denotes novel GWAS signal at time of submission.*

[c] *denotes signals which were reported by Bonazzola et al. (2024) and presented in Table 1 in their paper (UPE and summary statistics for the shape and reconstructed left ventricular GWAS).*

may have potential utility for CV risk prediction, especially those including novel loci (PCs 2, 3, 4 and 5).

While this manuscript was under review a paper was published by Bonazzola et al.[64], in which genetic analyses were performed on mesh-derived PCs of only the left ventricular shape at end-diastole using CMR data from the UK Biobank, alongside GWAS on autoencoder latent variable ensembles and autoencoder reconstructions of CMR measures. These models did not utilize valve locations from long-axis

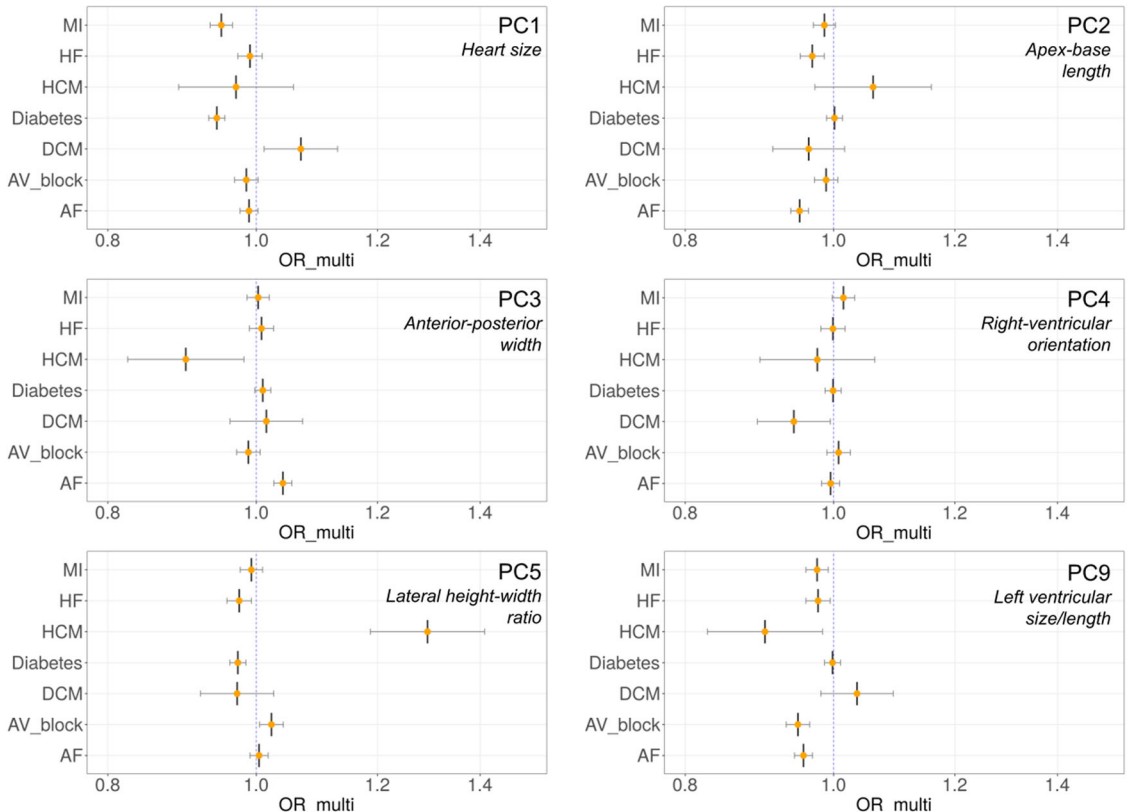

**Fig. 5 | Odds-ratio plots for relationship between PC PRS and cardiometabolic diseases.** Bar plots illustrating the odds ratios from the logistic regression of PC-derived PRS against the incidence of cardiometabolic diseases, after adjusting for relevant covariates ($n = 371,264$). A vertical blue dotted line is plotted at an odds-ratio of 1, with each individual disease having an odds-ratio plotted as a central vertical line and orange point, and confidence interval lines flanking as an intersecting horizontal line denoting 95% confidence intervals. PC, principal component; OR, odds ratio; PRS, polygenic risk score; MI, myocardial infarction; HF, heart failure; HCM, hypertrophic cardiomyopathy; Diabetes, diabetes mellitus; DCM, dilated cardiomyopathy; AV block, atrioventricular block; AF, atrial fibrillation; ICD codes used to define these diseases in Supplementary Data 28.

CMR images, and did not include long axis ventricular contours to guide mesh fitting, contrary to the current work in which long axis ventricular contours and valve landmarks (in particular including the aortic valve) were included from the automated image analysis of two- three- and four-chamber long axis cine images. Of the 45 genetic signals we report 19 were genome-wide significant ($P < 5 \times 10^{-8}$) in the results reported by Bonazzola et al.[64]. Additionally one variant (PC9:*NKX2-5*) demonstrated suggestive significance ($5 \times 10^{-8} < P < 1 \times 10^{-6}$), it was a variant in low LD ($r^2 = 0.19$) with the lead variant reported by Bonazzola et al. For the remaining 25 signals, none were in LD ($r^2 > 0.1$) with variants reported by Bonazolla et al. of these 22 had support ($P < 0.05$) and 3 signals had no support (PC1- *BAG3*, PC3 - *PRDM6*, and PC5 - *SIM2*) in the Bonazzola et al., dataset. The methodological differences likely explain the differences in the loci discovered. In Table 3 we indicate loci discovered by Bonazzola et al., which were genome-wide significant (either the lead variant or a high LD proxy $R^2 > 0.7$).

Our study has some limitations. Firstly, the shape PCs are derived from an unsupervised machine learning method; although we manually attributed biological features to the multi-dimensional variation observed in the PCs, these are high dimensional traits and they cannot be reduced to a simple anatomical interpretation. However, the PCs are readily obtained through well-understood image analysis and shape atlasing methods. They are also readily quantifiable by automatically analysing the images, customizing the shape model, and projecting this shape model onto the 11 PC modes to obtain the scores. This can be performed automatically at imaging, so the PC scores would be available as part of the imaging report. We also show that the PC's robustly characterize a significant amount of shape variation in non-European and disease cohorts, indicating that projection onto these PC modes can be robustly generalized to other cohorts. We also show that the PCs are interpretable with respect to clinical outcomes since they are both directly associated with cardiometabolic diseases, and the PRS associated with PCs are also associated with cardiometabolic diseases. This implies that the genetic factors underlying shape variation in the population are also involved in disease processes. A second limitation is that we restricted genome-wide association analyses to European ancestry individuals due to low sample size in other ancestries. Although we demonstrate the applicability of the European PRS to other ancestries with comparable performance and compare the results with a multi-ancestry PRS, a cross-ancestry GWAS would lack power due to the limited non-European UK Biobank imaging cohort.

The results from all our analyses will need to be validated and expanded, and the recruitment target of 100,000 individuals to be enrolled into the UK Biobank imaging study will permit this and extension of our findings across multi-ancestry individuals.

In the future, anatomical cardiac shape features informed by the PCs identified here can be further examined. The direction of effect between multidimensional heart shape and cardiometabolic disease should be evaluated with mechanistic studies, including further exploring the functionality of the identified candidate genes.

In summary, we report a genetic basis for heart shape PCs, as well as new information on cardiac shape biology. We demonstrate the potential for using genetically predicted shape PCs for prediction of adverse cardiometabolic diseases and suggest these could be used in tandem with existing PRS and clinical information to support CVD risk prediction and stratification.

## Methods

### UK Biobank

This project utilises data from the UK Biobank (application 2964); a large-scale prospective cohort study with 500,000 participants aged 40–69 with detailed health, lifestyle, physical measures and biological samples including genetic data (488,377 have genotype data collected across two genotyping arrays (UK BiLEVE, UK Biobank Axiom with 807,411/825,927 markers respectively, 95% overlap)). The study design, data collection methods and quality control steps have been described previously[65,66].

### Ethical approval

This study complies with the Declaration of Helsinki; the work was covered by the ethical approval for UK Biobank studies from the NHS National Research Ethics Service on 17th June 2011 (Ref 11/NW/0382) and extended on 18 June 2021 (Ref 21/NW/0157) with written informed consent obtained from all participants.

### Atlas-derived phenotypes

In UK Biobank CMR images were acquired on a clinical wide-bore 1.5 Tesla scanner (MAGNETOM Aera, Syngo Platform VD13A, Siemens Healthcare, Erlangen, Germany) utilising ECG-gating[67]. A full description of the UK Biobank CMR imaging protocol can be found here: https://doi.org/10.1186/s12968-016-0227-4.

Short and long axis cine images were analysed automatically using Circle cvi42 version 5.11 release 1505. The software used a deep learning convolutional neural network and returned LV endocardial, LV epicardial, and RV endocardial contours for short axis as well as three-chamber, two-chamber and four-chamber long axis views. Landmarks denoting the location of the mitral (all long axis views), tricuspid (four-chamber) and aortic (three-chamber) valves were extracted from the contour output.

A biventricular subdivision surface mesh model was customized to each case using a non-rigid diffeomorphic registration method described previously[32]. Briefly, slices were aligned to correct breath-hold misregistration and the template mesh was deformed iteratively to minimize the distance between contour points and their corresponding closest point projections in 3D.

The statistical shape atlas was constructed by aligning all the shape models using the Procrustes alignment method (translation and rotation only, scale was not removed). PCA was performed to derive principal components of shape variation across the cohort. Each principal component captures a unique and uncorrelated shape feature, and Z-scores per participant were used as phenotypes in GWAS for each principal component.

### Atlas quality control

A series of quality control steps was performed to detect and remove outliers and cases with incomplete or poor-quality data, summarised in Fig. 1. Firstly, cases that had an incomplete set of valve landmarks, or incomplete machine learning contours for 3D modelling, were excluded. Four types of shape-based quality scores were computed: 1) model-contour error in mm, being the average distance between the contour points and the closest model surface points; 2) individual principal component z-scores for the first 10 components; 3) Mahalanobis distance over the first 10 principal components, being the root sum of squares of the first 10 z-scores; 4) residual error in mm between the model and its projection onto the first 10 modes. In each case, scores 5 interquartile ranges above the lower quartile or below the upper quartile were excluded (full explanation in Supplementary Methods).

### Atlas and generalisability

To examine the robustness of the largely healthy European shape atlas in other groups, we calculated the amount of variance explained by the first 11 PCs in subsets of the study group. We calculated the percent variance explained by the first 11 PCs in Europeans ($n = 41,235$), non-Europeans ($n = 2441$); defined by the agreement of a participants self-reported ancestry at interview and genetically inferred ancestry[68], and in participants with previous myocardial infarction ($n = 671$), defined by ICD9/10 codes (Supplementary Data 28) and self-reported previous myocardial infarction. The calculation was for each group 100 x (the variation captured by the first 11 PCs / total variance explained).

### Genetics cohort quality control

From the 43,676 cases with models at end-diastole 39,449 passed the quality control ($n = 4227$ excluded) and were restricted to participants with an agreed self-reported and genetically inferred white European ancestry ($n = 2443$ excluded) due to the low non-European ancestry representation in the UK Biobank imaging study (defined using k-means clustering agreement with self-reported ancestry)[68]. To avoid the confounding effects of cardiovascular disease phenotypes the cohort was restricted LV ejection fraction $\geq 40\%$ (calculated in the 3D models, 1695 excluded) and no previously reported major adverse cardiovascular events (defined in Supplementary Methods) (917 excluded) utilising hospital episode statistics (HES) (Copyright © (2023), NHS Digital. Re-used with the permission of the NHS Digital [and/or UK Biobank]. All rights reserved). After removing cases with missing covariate data (91 excluded) and participants with discordant self-reported and genetically inferred sex, 35,055 remained in the analysis cohort (summarised in Supplementary Data 1).

### Correlation and association analysis

We used a two-sided Pearson correlation analysis in R (version 4.2.2)[69] to examine the relationships between our PCs and common CV structural and functional measures and risk factors in the atlas cohort. The CV risk factors examined included covariates used in the GWAS which are known to affect cardiac morphology, and smoking status. These measures are taken from the imaging visit of the UK Biobank's assessment. The structural and functional CV measures included well explored phenotypes studied in CMR GWAS; left and right end systolic and end diastolic volumes, ejection fractions and stroke volumes, myocardial masses and mass to volume ratios. The diseases were selected based on prior work indicating links between CV measures and diseases (heart failure, dilated and hypertrophic cardiomyopathy, atrial fibrillation, 2nd or 3rd degree atrioventricular (AV) block or pacemaker implantation, diabetes and myocardial infarction) (Supplementary Data 28). We used logistic regression to derive odds-ratios and confidence intervals for disease incidence. This regression was adjusted for covariates selected using stepwise regression to identify covariates significant for each outcome, selecting from: age, sex, height, BMI, alcohol consumption, adjusted systolic and diastolic blood pressures, smoking status, low density lipoprotein and mean triglyceride levels (Supplementary Data 3–9 (A)). We also assessed body surface area as an alternative covariate for body size: as a covariate BMI was found to perform comparably with body surface area in our phenotypes ($r^2$ between PC1 and height + BMI = 0.595, between PC1 and height + body surface area = 0.603. For testing associations between PCs and outcomes, the imaging visit data from the UK Biobank was used. Missing values were imputed for these covariates using MICE[70] in R, each imputed covariate had < 5% missing data. High density lipoprotein level was considered as a covariate and dropped as it had > 5% missing data.

### Genetic analyses

To maximise the power of the cohort all suitable participants ($n = 35,055$) were used as the discovery cohort. Variant quality control was performed using PLINK v1.9[71,72], applying filters of minor allele frequency > 1%, a Hardy-Weinberg exact test of $< 1 \times 10^{-6}$ and a missing

rate of < 1.5% to the SNVs, with a resulting 559,556 SNVs selected as model SNVs for the subsequent analyses.

The proportion of genetic variation explained by additive genetic variation (heritability) for each PC was estimated using a variance components method (BOLT-REML, version 2.4.1)[33] with a MAF threshold of ≥ 1% and INFO > 0.3, with model SNVs (defined above) and ~1.2 million imputed variants (using the UK10K reference panel[73]). BOLT-LMM[34] also computes the genetic correlations between included phenotypes, which we report in Supplementary Fig. 1.

Each PC was then analysed using a linear mixed model method (BOLT-LMM, version 2.4.1[34]) using the model SNVs and ~1.2 million imputed SNVs, as well as the heritability estimates obtained from BOLT-REML. Each model was regressed on genotype dosage using multiple linear regression with an additive genetic effect adjusted for height, age, sex, array type (UK Biobank/UK BiLEVE), systolic blood pressure (averaged between manual and automated readings and adjusted 15 mmHg in the presence of blood pressure altering medication), resting heart rate and body mass index, all variables which were significant in our correlation analysis (Fig. 3). These covariates were selected to adjust the mixed model for known traits that affect cardiovascular morphology, as is standard for cardiovascular genetic analyses in UK Biobank; height, sex and BMI are directly related to cardiac morphology. A $p < 5 \times 10^{-8}$ was used to declare genome-wide significance. LocusZoom[74] region plots for all significant loci identified are available in Supplementary Fig. 5.

To assess for confounding in our GWAS studies, the LD score regression intercept was estimated using LD score software (LDSC, v1.0.1)[75]. Genomic inflation factor was reported from the mean lambda value output by BOLT-LMM[34], and quantile-quantile plots were generated from the GWAS summary statistics with the FUMA (functional mapping and annotation of genome-wide association studies) functional annotation tool (version 1.4.0)[46].

In this study we define a genomic signal as the combination of a lead genome wide significant SNV and all SNVs within a 1mb window in LD $r^2 > 0.1$ with that lead. In contrast we define a locus as the 1mb window around that lead variant, disregarding LD.

## Conditional analysis

Conditional analysis was performed using GCTA (version 1.94.1)[35]. This analysis was applied to all genome-wide significant loci identified in the GWAS. Mirroring conservative methods described previously[76], a secondary signal would be declared if the newly identified SNV original $p$-value was lower than $1 \times 10^{-6}$, if there was less than a 1.5-fold difference between the lead SNV and secondary association $p$-values on a $-\log 10$ scale, i.e., if $-\log_{10}(P_{lead})/-\log_{10}(P_{sec}) < 1.5$, or if there was less than a 1.5-fold difference between the main association and conditional association $p$-values on a $-\log_{10}$ scale, i.e., if $-\log_{10}(P_{sec})/-\log_{10}(P_{cond}) < 1.5$.

## Percent variance

The percent variance explained for each PC by all genome-wide significant variants and independent secondary signals was calculated using standard regression models, including the covariates from the GWAS (described above). Each phenotype was regressed on the analysis covariates, and the $r^2$ value was used as the estimation for percent variance explained[77].

## Signal cross-lookup

We wanted to examine the significance of our genome wide significant SNVs across other PCs, to assess the effects of our shape signals amongst independent derivations of shape. For our lead SNVs and conditionally independent secondary signals we perform a lookup of that SNV and its high LD proxies ($r^2 > 0.8$) in our GWAS summary statistics, and for each PC report the most significant $p$-value from the

lead and proxies in those summary statistics, summarised in Supplementary Data 10.

## Gene-based annotation study

Genome-wide gene-based associations were assessed using the Multi-marker Analysis of Genomic annotation tool[36] (MAGMA v1.06) from the FUMA GWAS online tool[46]. Variants in the raw genotype summary statistics were assigned to genes based on the overlap of their genomic location with genes within a window of 35kb upstream and 10kb downstream to include regulatory elements. The MAGMA tool computes a gene-based $p$-value for the protein coding genes mapped to their assigned SNVs. We report genes with a computed $p$-value of $2.6 \leq 1 \times 10^{-6}$ (Bonferroni corrected 0.05/19,414 genes tested).

## Bioinformatics analyses

**Variant level annotation.** Analyses were performed to annotate the identified lead SNVs and their proxies ($r^2 \geq 0.8$), and the secondary signals. Positional information and GWAS summary statistics were extracted, and each lead or conditionally independent SNV was assessed with Ensembl Variant Effect Predictor (release 105.0)[37], a collation of tools used for assessing and predicting the effects of SNVs and their impacts on human biology (through SIFT version 5.2.2[33] and PolyPhen-2 version 2.2.2, release 405c[39]).

To assess for potential effects of lead and conditionally independent variants (and their proxies [$r^2 \geq 0.8$]) on tissue specific gene expression in LV, aortic artery, coronary artery and atrial appendage tissues, they were first checked for overlap with lead eQTL variants at each tissue using the GTEx (version 8) database[40,78]. Subsequently, colocalization analyses were performed using the COLOC package (version 5.1.0.1)[41] in R to analyse each eQTL-GWAS dataset pair. This tool uses Bayesian statistical methodology to test the pairwise colocalization of SNVs in a GWAS with eQTLs, and generates posterior probabilities for each locus, weighting the evidence for competing hypotheses of no evidence of colocalization or the sharing of a distinct SNV at each locus. We used a posterior probability (PP) of PP4 > 0.75[41] to indicate strong evidence of a tissue-specific eQTL-GWAS pair influencing both the expression and GWAS trait at a particular region for the specified GTEx tissues of interest.

Variants high LD proxies ($r^2 > 0.8$) were also assessed for their regulatory potential using RegulomeDB[43], to find genes whose promoter regions form significant chromatin interaction with these SNVs in left and right ventricular and aortic tissues. This utilises long-range chromatin interaction (Hi-C) data from FUMA[79] and Jung[80] datasets, which use different methods and resolutions of Hi-C data and provide additional discovery power when used together. Target genes were only considered with evidence of significant enhancer-promoter interactions at FDR $< 1 \times 10^{-6}$, and filtered to regulatory GWAS variants with a RegulomeDB[35] score of < 5 (where a lower score indicates greater evidence of functional significance) that were in LD ($r^2 \geq 0.8$) with our lead SNVs, and chose the interactors of highest regulatory potential to annotate the loci.

## Gene-level annotation

**Transcriptome-wide association study.** We also employed the PRE-DIXCAN tool (version 0.6.5)[42], a gene-based association method that tests the molecular mechanisms through which genetic variation affects a phenotype, to perform a TWAS to predict the effects of gene expression levels on each of our PCs. We used S-PREDIXCAN; an extension of the original PREDIXCAN tool that infers results from GWAS summary statistics and alleviates the need for individual-level genotype or phenotype data. S-PrediXcan provides a precalculated transcriptome model database from GTEX-based tissues and covariance matrices of SNVs within each gene model (https://github.com/hakyimlab/MetaXcan). A Bonferroni corrected threshold (0.05/

number of tissue-gene pairs tested $0.05/16,097 = 3.1 \times 10^{-6}$) was used to declare significant results.

**Evidence for candidate genes.** We generate a list of genes at the locus (genes within 50kb of the lead SNV and at $r^2$ of > 0.5) of each lead and secondary SNV. Candidate genes were prioritised at each locus if they had two or more supporting lines of evidence from the following criteria; presence of missense variant at the locus, gene prioritised by S-PREDIXCAN or eQTL analysis, availability of knockout model from International Mouse Phenotyping Consortium (http://www.mousephenotype.org) and the Mouse Genome Informatics database[45] (http://www.informatics.jax.org/) with a CV phenotype, support from literature review on function of the gene with CV disease or Mendelian CV disorder, target gene from Hi-C data from FUMA[79] and Jung[80] datasets, the gene being located within the 100kb window of the lead variant (nearest gene) or the gene being prioritised by MAGMA.

**Functional enrichment.** GARFIELD[44] (version 2) was used to identify tissue-specific enrichment of variants at DNase I hypersensitivity sites in each of our PCs. GARFIELD annotates variants with data from the ENCODE, GENCODE and Roadmap Epigenomics projects and calculates odd ratios using a generalised linear model framework. GARFIELD reads variants from our GWAS summary statistics using two $p$-value thresholds: genome-wide significance ($p = 5 \times 10^{-8}$) and borderline significant ($p = 1 \times 10^{-6}$) in its analysis.

**Pathway analysis.** Our candidate genes were queried in the Gene2-Func pathway analysis tool from FUMA[46] to perform functional enrichment analysis and identify significantly associated gene ontology (GO) terms and biological pathways from Kyoto Encyclopaedia of Genes and Genomes (KEGG), Reactome and WikiPathways and results from GWAS catalog among others (Supplementary Data 19).

**Literature lookup.** We also checked which of our SNVs are not previously reported in CMR-derived phenotype GWAS through literature review and lookup tools GWAS Catalog[47] and Phenoscanner[48,49] (Supplementary Data 21, 22). For each genome wide significant SNV in our study we consider both the lead SNV and its high linkage proxies ($r^2 > 0.8$), searching these in GWAS Catalog and Phenoscanner. We report all significant ($p = 5 \times 10^{-8}$) GWAS associations, focusing on those that are associated with the cardiovascular system (cardiac structural and functional measures, ECG measures, cardiac disease traits) for novelty assessment. We have also collated through literature review information on previously reported cardiovascular trait loci, from results tables and summary statistics where available. Using this, we again performed a lookup of our lead variants and proxies for their association with CV traits, and if any of these variants fall within 500 kb of a previously reported CV SNV, reporting if the previously reported cardiac trait SNV is in linkage disequilibrium $r^2 \geq 0.1$ with our SNV.

**PheWAS.** To identify evidence of pleiotropy with clinical conditions we implemented a phenome-wide association study (PheWAS) using the R package PheWAS[50] (version 0.99.5-5). ICD9 and ICD10 codes from the UK Biobank HES data and mapped to phecodes. Lead and conditionally independent secondary variants from unrelated (kinship coefficient > 0.0884) white-European UK Biobank individuals not included in the GWAS ($n = 389,449$) were tested for association with these phecodes, adjusted for age, sex and the first 10 genetic principal components. In the output a Bonferroni corrected threshold (for the number of phecodes tested with $\geq 200$; $0.05/325 = 1.54 \times 10^{-4}$) was used to declare significance.

**Polygenic risk scores.** To determine relationships between PC associated genetic loci and cardiometabolic diseases, PRS were constructed using the lead variants from PCs 1-5 and PC9 from a UK Biobank unrelated European cohort (kinship pairs < 0.0884) not included in the GWAS dataset with concordant self-reported and genetically inferred sex ($n = 371,264$). We tested for associations with heart failure, myocardial infarction, atrial fibrillation, hypertrophic and dilated cardiomyopathies, 2nd or 3rd degree atrioventricular block or pacemaker implantation, and diabetes (Supplementary Data 28), the same diseases that we include in our correlation analyses. Utilising genotype probability data in the BGEN format we use PRSice-2[51] (version 2.3.5) to calculate PRS, summing the dosage of each shape allele weighted by effect size (from GWAS). Associations are calculated through logistic regression of standard deviation increase in PRS against cardiometabolic disease incidence, including covariates selected to be significant for each disease through stepwise regression (see "method" in *Correlation and association analysis*, Supplementary Data 3–9 (B)), alongside genetic array used and the first 15 genetic principal components. For these regressions the first visit UK Biobank covariate data was used. A $p$-value threshold of < 0.0071, Bonferroni corrected was chosen (0.05/number of diseases)

We subsequently examined the same PRS in a multi-ancestry UK Biobank cohort with the same kinship restrictions ($n = 388,152$) and covariates. The same $p$-value threshold was used as in the European PRS analysis. Ancestries were defined as agreement between self-reported ethnic background and genetically inferred ancestry (k-means clustering)[68].

To validate the application of the European PRS in non-European populations we first investigated the effect allele frequency of the European GWAS lead variants in African ($n = 6716$), East Asian ($n = 1671$) and South-East Asian ($n = 8501$) populations (defined above in *Atlas and generalisability*, as well as the allele frequency combined across ancestries (Supplementary Data 25). This comparison was performed using both the UK10K reference panel[73] which was used for the imputation of variants for our GWAS and the 1000 Genomes phase 3 reference panel to ensure there were no major discrepancies between reference panels.

We also examined the percentage variance explained by our PRS in the non-European ancestries by regressing respective PC scores against PRS and the covariates used in GWAS (Supplementary Data 26) in each individual ancestry. To do so we utilised the non-European individuals of the UK Biobank's CMR imaging cohort who were included in the construction of the atlas (African = 225, East Asian = 140, South-East Asian = 424).

To validate the association between the PC5 PRS and HCM, we used an external HCM cohort, comprising individuals with HCM referred to the Inherited Cardiovascular Disease unit at St. Bartholomew's Hospital, London, UK, the Inherited Cardiovascular Disease Unit at The Heart Hospital, University College London Hospitals (UCLH), London, UK and the Unidad de Cardiopatías Familiares of Complexo Hospitalario Universitario A Coruña, Spain. All patients gave written informed consent for genetic testing, and the study was approved by the regional ethics committees (London: 15/LO/0549; Coruna: 2021/182). Clinical parameters were recorded using previously described methods and were stored in a dedicated database[81]. HCM was defined by the presence of a maximal left ventricular wall thickness $\geq 15$ mm in probands or $\geq 13$mm in relatives. Patients with previously confirmed HCM phenocopies (e.g., Fabry disease, amyloidosis, glycogen storage diseases, and RASopathies) were excluded from the study. The samples used in this study were collected from 2011 to 2019 in London and 1991 to 2021 in Coruna. The HCM samples from the two centres ($n = 3024$) were genotyped at University College London Genomics unit using the Infinium Global Screening Array-24SA v3.0 with a standard methodology.

Quality-control checks on the genotype data were performed using PLINK v1.9[71,72]. Individuals were excluded with a genotype missing rate of > 5%, heterozygosity exceeding three standard deviations

from the mean or inferred sex discordance. At a variant level, variants were excluded if exceeding a Hardy-Weinberg equilibrium of $1 \times 10^{-6}$ or missingness > 2% (for variants with a minor allele frequency of ≤ 3%, a more stringent missing rate cut-off of ≥ 1% was applied). Patients were checked for relatedness using ~600,000 variants using the KING relationship inference tool v2.2.6[82] excluding up to and including second-degree relatives.

We constructed the PC5 PRS in 2,284 HCM patients from this dataset, and 40,825 European case controls from the UK Biobank, matched by age (decade) and sex, these individuals were not included in the UK Biobank's imaging cohort, excluding individuals carrying rare variants (MAF < 0.001) in the eight most established sarcomeric genes for HCM (*MYH7, MYBPC3, TNNT2, TNNI2, ACTC1, MYL2, MYL3, TPM1*) from the whole-exome sequencing data available in ~250,000 individuals at analysis. Our PRS included the lead SNVs from the PC5 GWAS, however of the 9 SNVs used in the PRS two were not present in the genetic data from the HCM cohort, therefore proxies in high LD were used instead. These were rs10155223 and rs146170154, substituted respectively with rs1500800 ($r^2 = 0.98$) and rs3176326 ($r^2 = 0.99$). We regressed the standardised PRS against case-control status.

### Mendelian Randomization

The TwoSampleMR[83] R package (version 0.6.6) was used to test for association of PC1 and T2D, as well as of PCs 3 and 5 and AF, the strongest associations found from observational associations. Summary statistics for T2D from Loh et al.[84] (19,291 T2D cases, 440,033 controls) and for AF data from Nielsen et al.[85] (60,620 AF cases, 970,216 controls) were downloaded from the GWAS catalog. For all MR analyses, lead SNVs ($P < 5 \times 10^{-8}$) were taken as genetic instruments. Effect alleles were harmonised between IVs and summary statistics. Three variants associated with PC1 were not present in the T2D summary statistics, and for each of these the highest LD proxy variant present in the summary statistics was selected: rs7773935 (chosen proxy: rs1634754, $r^2 = 0.88$), 10:121347839:TTTTC:T (chosen proxy: rs3816145, $r^2 = 0.96$), rs533030436 (chosen proxy: rs76774446, $r^2 = 0.91$). To account for weak-instrument bias, the F-statistic was calculated for each lead SNV, and only those with an F > 10 were kept as instrumental variables. SNV's were subsequently pruned using the ieugwasr package[86] in R (version 1.0.0) to remove correlated variants ($r^2$ 0.001) within 10Mb, preserving the lowest P-value variant for association analyses. Four Mendelian randomization methods were used for this analysis: IVW, weighted median, MR-Egger and MR-PRESSO. Results are reported as OR (95% CI) when testing for a causal association with T2D or AF, and as effect size (95% CI) when testing for a causal association with PC1, PC3 or PC5.

### Reporting summary

Further information on research design is available in the Nature Portfolio Reporting Summary linked to this article.

## Data availability

All UK Biobank data are available upon application (www.ukbiobank.ac.uk). Data in this study was accessed on application 2964. Summary GWAS statistics are publicly available on the GWAS catalog portal (https://www.ebi.ac.uk/gwas/) through study accessions: GCST90448480, GCST90448481, GCST90448482, GCST90448483, GCST90448484, GCST90448485, GCST90448486, GCST90448487, GCST90448488, GCST90448489 and GCST90448490.

## Code availability

All code used to generate models, the shape atlas and subsequent analyses is available upon request.

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

## Acknowledgements

This research was conducted using the UK Biobank Resource under Application Number 2964. The work uses data provided by patients and collected by the NHS as part of their care and support. This research used data assets made available by National Safe Haven as part of the Data and Connectivity National Core Study, led by Health Data Research UK in partnership with the Office for National Statistics and funded by UK Research and Innovation (grant ref MC_PC_20029). Copyright © (2022), NHS Digital. Re-used with the permission of the NHS Digital [and/or UK Biobank]. All rights reserved. Barts Charity (G-002346) contributed to fees required to access UK Biobank data [access application #2964]. RB and AY acknowledge core funding from the Wellcome/EPSRC Centre for Medical Engineering [WT203148/Z/16/Z] and The London Medical Imaging & AI Centre for Value Based Healthcare. RB is funded by the EPSRC Research Council, part of the EPSRC DTP, Grant Ref: EP/R513064/1. AY is supported by National Institutes of Health R01HL121754. WJY was supported by an MRC CRTF grant MR/R017468/1. WJY acknowledges the NIHR Integrated Academic Training programme, which support their Academic Clinical Lectureship posts. NA acknowledges the Medical Research Council (MRC) for his Clinician-Scientist Award (MR/X020924/1). JR acknowledges funding from the European Union's Horizon 2020 Research and Innovation Programme under the Marie Sklodowska-Curie grant agreement number 786833, from the European Union-NextGenerationEU and fellowship RYC2021-031413-I from MCIN/AEI/10.13039/501100011033 and from the European Union "NextGenerationEU/PRTR". SEP acknowledges the British Heart Foundation for funding the manual analysis to create a cardiovascular magnetic resonance imaging reference standard for the UK Biobank imaging resource in 5000 CMR scans (www.bhf.org.uk; PG/14/89/31194), as well as from the "SmartHeart" EPSRC programme grant (www.nihr.ac.uk; EP/P001009/1). SEP has received funding from the European Union's Horizon 2020 research and innovation programme under grant agreement No 825903 (euCanSHare project). SEP and PBM acknowledge the support of the National Institute for Health and Care Research Barts Biomedical Research Centre (NIHR203330); a delivery partnership of Barts Health NHS Trust, Queen Mary University of London, St George's University Hospitals NHS Foundation Trust and St George's University of London. LRL and the UCL HCM cohort genotyping were supported by an MRC clinical academic partnership award MR/T005181/1. LRL, PS and PE acknowledge funding from the NIHR UCLH Biomedical Research Centre Cardiovascular Diseases Theme (project no. BRC767).

This article is also supported by the London Medical Imaging and Artificial Intelligence Centre for Value Based Healthcare (AI4VBH), which is funded from the Data to Early Diagnosis and Precision Medicine strand of the government's Industrial Strategy Challenge Fund, managed and delivered by Innovate UK on behalf of UK Research and Innovation (UKRI). Views expressed are those of the authors and not necessarily those of the AI4VBH Consortium members, the NHS, Innovate UK, or UKRI. This work was supported by Health Data Research UK, an initiative funded by UK Research and Innovation, Department of Health, and Social Care (England) and the devolved administrations and leading medical research charities. The funders provided support in the form of salaries for authors as detailed above but did not have any additional role in the study design, data collection and analysis, decision to publish, or preparation of the manuscript.

## Author contributions

W.J.Y., J.R., A.Y., and P.B.M. conceived and designed the experiments, R.B., W.J.Y., C.S., N.A. performed the experiments, R.B., W.J.Y., C.S. performed statistical analysis, R.B., W.J.Y., J.R., A.Y., and P.B.M. analysed the data, W.J.Y., N.A., L.R.L., P.M.E., P.S., P.E., R.B.V., J.R., A.Y., and P.B.M. contributed reagents/materials/analysis tools, J.R., A.Y., and P.B.M. jointly supervised the research, R.B., W.Y., J.R., A.Y., and P.B.M. wrote the paper and N.A., L.R.L., P.M.E., P.S., and S.E.P. provided critical review of the manuscript.

## Competing interests

Dr Barriales-Villa has received consultant/advisor fees from MyoKardia/Bristol Myers Squibb, Cytokinetics, Sanofi, Pfizer, Amicus and Alnaylam. The remaining authors declare no competing interests.
