## [Peer Review File · Nature Communications]

Genetic basis of right and left ventricular heart shapeREVIEWER COMMENTS

Reviewer #1 (Remarks to the Author):

Burns et al. present a well-written and comprehensive analysis of the UK Biobank, assessing novel derived ventricular shape features, their genetic correlates, and the subsequent associations of both the features and genetic correlates. Briefly, they take approximately 45K UKBB participants with cardiac MRI, selecting for preserved ejection fraction, white ethnicity, lack of cardiovascular disease, and quality control, resulting in around 35K cases which are contoured to obtain end diastolic mesh models. These models are analyzed using principal component analysis to identify the 11 primary principal component shapes. These shapes are analyzed for associations with traditional measures and incident future cardiometabolic disease. These components were analyzed with multiple approaches for heritability and specific loci associated with each component, resulting in significant findings in 43 loci, 45 signals, in 8 components. The genetic variants were tested for association with disease processes, organ relations, interactions, biological processes, and associations with cardiac traits.

The genetic findings for the shape components were then combined into polygenic risk scores, and expanded into analysis within the UKBB of patients without CMR, but with genetic testing and subsequent outcomes. These approximately 390K patients were assessed for the association between the polygenic scores and cardiovascular outcomes, not limited to the original white-only derivation cohort, but still within the UKBB.

Through these steps, they establish different anatomical shape components, their genetic basis, cardiovascular significance, and potential genetic mechanistic factors in a large cohort using cardiac MRI. With the caveat that as a reviewer I lack individual experience with many of the genomic analysis packages leveraged for this work and therefore cannot comment directly on the application using these packages, I felt that this was an exceptionally well-written, comprehensive, and interesting manuscript. It leverages novel approaches and demonstrates the immense potential of the data in the UK Biobank, by revealing new conceptual frameworks and insights for genetic basis of heart shape which are novel and clinically meaningful. It was a pleasure to read and I found no major or minor errors or corrections to suggest, and believe it should be accepted without further revision.

Reviewer #2 (Remarks to the Author):

Analyzing participants of UK Biobank with cardiac MRI and genotyping data the authors performed a GWAS on heart shape. After identification of distinct patterns of cardiac shape by principle component (PC) analysis, 45 significant associations were identified by GWAS which were further explored by bioinformatics tools in order to narrow down potential genes and mechanisms underlying the associations. The authors are experts in the field and the paper is written very well.

While the authors indicate that their study is the first to examine the genetic architecture of heart shape one may specify: as measured by principle component (PC) analysis of MRI based images analyzed by a previously reported algorithm. Indeed, there are multiple GWAS studies on cardiac structural phenotypes and thus clinically established measures of heart shape. Accordingly, most genetic associations have been reported before for such cardiac traits.

The authors mention some limitations of their study, but there may be more that need consideration.

1.) It is unclear to this reviewer whether the analysis might be reproducible on other MRI data sets. A non-supervised clustering may indeed create PCs that are specific to a specific data set. Thus, any

validation of the findings may be challenging – and, accordingly, was not part of the manuscript.

Along the same lines, the clinical value of the observations may be enhanced if specific anatomical features of the PCs (like shown in table 1) are further explored. Rather than the multifactorial (and intransparent) PCs such measures may be put into focus in further studies. A potentially interesting observation in this regard is a component of PC2, i.e. "Orientation of the valves and the length to lateral dimension ratio" which associates with "AF, AV block". Such traits and respective associations may be validated and lead to biological insights with relevance for human biology.

2.) It is unclear, whether the PCs were adjusted for body size. Obviously, larger people have larger hearts such that PCs may be indirectly affected by non-cardiac traits. Indeed, PC1 (which explained most of the variability) was strongly correlated with height and inversely with age. Thus, it is unclear whether this PC is rather a function of body composition than a specific feature of the heart.

In this respect it would be interesting to learn whether some of 14 signals not previously reported for any cardiac trait, of which, however, 7 showed associations with non-cardiac traits, display association with measures of body size and or body composition. In fact, according to the authors only 7 loci have not been previously reported in any GWAS.

3.) The authors relate PCs to traditional anatomical features, i.e. descriptively, PC1 was associated with overall heart size, PC2 with apex-base length and sphericity – ect. However, it is unclear whether the (genetic) associations with the PCs also apply to these anatomical features, since the PCs explain only little of their variability.

4.) Of the 45 loci with significant association to any of the PCs, only 6 were not associated with any other PC. Thus, it may be questioned as to whether the PCs reflect unique traits or have a strong overlap.

5.) The discussion on the potential functional effects of the affected genes is highly speculative since association with a "PC" rather than an anatomically quantifiable measure precludes any understanding what the genetic variant may do.

6.) Directionality/Causality of associations are unclear. For example, some PCs of heart shape associate with diabetes mellitus. It is unclear whether the genetic/anthropometric underpinnings of the PC also affect risk of DM – or whether the subsequent manifestation of DM affects the probability to present with a specific PC.

7.) The authors may consider to mention that the associations observed for PCs by GWAS were explored bioinformatically but not validated experimentally.

8.) It is also challenging to explain/understand why PC5 and its PRSs display opposing directionality with regard to risk of HCM.

Reviewer #3 (Remarks to the Author):

Thank you for the opportunity to review the manuscript, "Genetic basis of right and left ventricular heart shape," by Burns and colleagues. This work aims to define and study the principal components of variation in cardiac morphology, as defined by surface meshes derived from MRI data. This approach is novel and has the potential to offer unique insights distinct from what may be gathered

from studying traditional MRI measurements. However, I think there are some areas where the manuscript requires additional details or improvement. My comments are below.

MAJOR

1. Although many aspects of this paper are clearly novel, I do not think the authors can claim to be the first to study the genet basis of heart shape. A published paper by Vukadinovic et al. (PMID: 36996817) analyzed cardiac sphericity, including GWAS. That paper should probably be cited, given the authors here spend significant text discussing cardiac sphericity as an important feature captured by their PCs. Another group (Meyer et al. PMID: 32814899) has studied ventricular fractal dimensions. Though, perhaps fractal dimensionality is distinct from shape, which is a somewhat non-specific term. In any case, I do not think the authors can claim to be the first to study the genetics of heart shape.

2. One challenge of reading this paper is understanding what the PCs represent. Of course, determining the nature of PCs derived from high-dimensional data is not an easy task and requires some subjective human interpretation. However, I think the authors should provide more explanation, data, or examples to better help the reader understand each PC. The GIFs are helpful, but they are not labeled, so it is unclear which GIFs represent which PCs. Further, there should be better consistency in the manuscript. For example, Figure 2 describes PC8 simply as "LV morphology." But Table 1 describes PC8 as "Tricuspid opening size and LV wall thickness." PC2 is described in the text as representing "roundness" but in Figure 2 it is labeled as "Apex-base length", and Table 1 suggests that it represents "orientation of the valves." The text also describes PC2, PC3, PC4, and PC5 as representing sphericity. The difficulty in knowing and describing what the PCs really represent may be a limitation of this method.

3. Related to the above, since defining the PCs is central to the paper, I think it is important that all PCs are included in Figure 2. Also, why is PC11 "not assessed" in Table 1? It is also not included in Figure 3. If PC11 is not going to be analyzed whatsoever, then perhaps it should be excluded from the paper all together.

4. The authors choose to only show PCs 1-5 and PC9 in Figure 4. They note, "In this association testing we selected only PCs with promising GWAS results for comparison with PRS associations - PCs 6, 7 and 11 had no genome-wide significant loci ($p < 5 \times 10^{-8}$) and were dropped, and PCs 8 and 10 had limited loci for PRS construction (< 3 SNV's) so were also dropped." This section is a bit confusing because at this point in the manuscript, no GWAS results have been presented. If the genetic analyses motivate which PCs to use for outcomes analyses, then I would recommend placing the genetic analyses first.

5. The methods section for the logistic regression testing the association between PCs and outcomes is not detailed. What covariates were included? Do these results represent a multivariable model with all PCs included (ie are the odds ratios the independent effects of each PC)? Or this simply a univariable analysis of each PC versus each outcome? If the results in Figure 4 only show univariable analyses, then it would be useful to also describe in text or show in a figure which PCs independently predict which diseases.

6. OR plots should use a log scale for the X-axis and be centered on 1. Otherwise, protective effects appear weaker than risk effects despite having the same magnitude of beta.

7. Subjects with prevalent MACE were excluded. How as MACE defined and derived?

MINOR:

1. The authors described GTCA analysis, writing, "Conditional analyses with genome-wide complex trait analysis (Methods) indicated two independent signals..." I believe they mean to say, "two additional independent signals."

Genetic basis of right and left ventricular heart shape - Response to Reviewers

The authors would like to thank the Reviewers for their rigorous and insightful review of the research. These comments are important for the interpretability of the methodology and its clinical implications. We have addressed the comments in order, and highlight the relevant changes made in the manuscript. The comments are numbered by both Reviewer and comment, (e.g., R1.1 for Reviewer 1, point 1, etc) and labelled in highlighted changes in the manuscript. In this document, we highlight in yellow insertions into the text, and relevant removals from the text are crossed through.

Firstly, we provide a summary of the main changes to the manuscript.

Introduction

- We have added a section to the end of the introduction to discuss the results of the study in compliance with the formatting instructions.

Methods and Results

- We have added PC11 to figures and tables (Figures 2, 3a, 3b, and 4, Supplementary movies 31-33) in response to a query from Reviewer 3.
- We have harmonised descriptions of the shapes represented by the PCs as requested. We now use the same descriptions throughout the paper and provide a full in-depth description in the Supplementary. In describing the shapes in results, we also change our description of which PCs relate to sphericity as this led to some confusion amongst reviewers about what the shapes were.
- Results are now presented for all PCs with risk factors, structural traits and cardiometabolic diseases.
- We revisited our analysis on testing for association with risk factors and cardiometabolic diseases. We are now systematic in how we selected covariates. We used a stepwise regression method to select covariates which are significant for each of the cardiometabolic outcomes. After implementation of this change, the associations between PC score/PRS and disease incidence changed a little, so the numbers have been updated. The methods and results tables have all been updated with the new results.
- There is a difference in the number of individuals included in the regression analyses from the prior submission:
 - In testing the PRS in European participants we previously erroneously did not exclude individuals that were included in the atlas, this has now been rectified. We now test the PRS in 371,264 participants (previously 389,449) of European ancestry.
 - The numbers of individuals of non-European ancestries used to test the PRS in a multi-ancestry cohort also differs. We previously used the self-reported numbers for UK Biobank participants, however in the revision we use an agreement of self-reported ancestry and genetically inferred ancestry. The multi-ancestry cohort for testing the PRS now includes: African cohort 6,716 (previously 16,012), South Asian 8,501 (previously 17,822) and East Asian

1,671 (previously 10,623). The methods and results are updated to reflect this change.

- Reviewer 2 requested follow up on the interpretation of the discordance of PC5 with HCM between the observational and PRS results. To investigate this we used an external cohort of 2,284 European HCM patients with age and sex matched UK Biobank controls. The results of this new analysis are now incorporated in the paper, with new text added in methods and discussion.

Discussion

- Several of the comments from reviewers have been incorporated into limitations and future work sections of the discussion.

Supplementary

- Supplementary Tables have been changed to reflect changes in methods of regressions, now detailing per outcome stepwise regression for covariate selection, observational associations between PC score and outcomes, and genetic associations between PRS and outcomes.
- Supplementary Methods now describes our definition of major adverse cardiovascular events (MACE) using ICD codes, and details in depth our assessment of the shapes captured by the PCs.
- The GIF representations of the shape PCs in different orientations have been renamed to supplementary movies in line with Nature Communications supplementary requirements.

Reviewer #1 (Remarks to the Author):

Burns et al. present a well-written and comprehensive analysis of the UK Biobank, assessing novel derived ventricular shape features, their genetic correlates, and the subsequent associations of both the features and genetic correlates. Briefly, they take approximately 45K UKBB participants with cardiac MRI, selecting for preserved ejection fraction, white ethnicity, lack of cardiovascular disease, and quality control, resulting in around 35K cases which are contoured to obtain end diastolic mesh models. These models are analyzed using principal component analysis to identify the 11 primary principal component shapes. These shapes are analyzed for associations with traditional measures and incident future cardiometabolic disease. These components were analyzed with multiple approaches for heritability and specific loci associated with each component, resulting in significant findings in 43 loci, 45 signals, in 8 components. The genetic variants were tested for association with disease processes, organ relations, interactions, biological processes, and associations with cardiac traits.

The genetic findings for the shape components were then combined into polygenic risk scores, and expanded into analysis within the UKBB of patients without CMR, but with genetic testing and subsequent outcomes. These approximately 390K patients were assessed for the association between the polygenic scores and cardiovascular outcomes, not limited to the original white-only derivation cohort, but still within the UKBB.

Through these steps, they establish different anatomical shape components, their genetic basis, cardiovascular significance, and potential genetic mechanistic factors in a large cohort using cardiac MRI. With the caveat that as a reviewer I lack individual experience with many of the genomic analysis packages leveraged for this work and therefore cannot comment directly on the application using these packages, I felt that this was an exceptionally well-written, comprehensive, and interesting manuscript. It leverages novel approaches and demonstrates the immense potential of the data in the UK Biobank, by revealing new conceptual frameworks and insights for genetic basis of heart shape which are novel and clinically meaningful. It was a pleasure to read and I found no major or minor errors or corrections to suggest, and believe it should be accepted without further revision.

Response:

The authors thank the Reviewer for their favourable review of the work and are grateful for their kind words on both its writing and methodology.

Reviewer #2 (Remarks to the Author):

Analyzing participants of UK Biobank with cardiac MRI and genotyping data the authors performed a GWAS on heart shape. After identification of distinct patterns of cardiac shape by principle component (PC) analysis, 45 significant associations were identified by GWAS which were further explored by bioinformatics tools in order to narrow down potential genes and mechanisms underlying the associations. The authors are experts in the field and the paper is written very well.

R 2.1 - While the authors indicate that their study is the first to examine the genetic architecture of heart shape one may specify: as measured by principle component (PC) analysis of MRI based images analyzed by a previously reported algorithm. Indeed,

there are multiple GWAS studies on cardiac structural phenotypes and thus clinically established measures of heart shape. Accordingly, most genetic associations have been reported before for such cardiac traits.

The authors mention some limitations of their study, but there may be more that need consideration.

Response:

We have modified the Abstract and Discussion sections to clarify that this is the first study of multidimensional shape features beyond traditional structural measures of mass, volume, and dimension ratios. The manuscript has also been modified to reflect the novelty of the work more accurately.

The limitations in the Discussion section have been updated in light of the comments made by the Reviewers, and now includes a more complete description of limitations in the project and how these are planned to be addressed in future work.

Abstract

*Heart shape captures variation in cardiac structure beyond traditional phenotypes of mass and volume. Although observational studies have demonstrated associations with cardiometabolic risk factors and diseases, its genetic basis ~~has not been investigated~~ **is less understood**. (Line 32)*

*Our study explores the genetic basis of **multidimensional** heart shape **using PCA** ~~for the first time~~, reporting new loci and biology, as well as polygenic risk scores for exploring genetic relationships of heart shape with cardiometabolic diseases. (Line 40)*

Introduction

***In this work, we discover there is a significant genetic contribution to PC-derived multidimensional cardiac shape phenotypes and demonstrate their associations with several prominent cardiometabolic diseases.** (Line 82)*

Discussion

*This is the first study to examine the genetic architecture of **multidimensional** ~~model-derived~~ heart shape **using PCA**. (Line 286)*

R 2.2 - It is unclear to this Reviewer whether the analysis might be reproducible on other MRI data sets. A non-supervised clustering may indeed create PCs that are specific to a specific data set. Thus, any validation of the findings may be challenging – and, accordingly, was not part of the manuscript.

Response:

We agree that the identified PCs are dependent on the dataset, and require validation in an external cohort of comparable size and imaging technique. As stated in the limitations, we were not able to perform external validation because we do not have access to such a dataset. However, the large cohort size (~40k) and limited number of PCs (11) studied in this work gives us confidence that these PCs are robustly characterized and would be generalizable to other cohorts.

Comparison with another cohort is more typically performed by projecting the new heart shapes onto the existing PC modes to derive the new shape scores with respect to the existing

atlas, rather than deriving new PC modes. In this approach the question becomes how well the existing PC capture shape variation in the new cohort. As a surrogate test of reproducibility in other cohorts, we have computed the % variance explained in heart shape by our PCs in non-European and MI sub-cohorts, to demonstrate the degree to which the 11 PCs capture heart shape in other cohorts. The first 11 PCs captured 83.6% of the variation over the entire cohort. Considering just the non-European individuals (n=2,441), the variance explained by the first 11 PCs was 83.0%, only slightly less than the European-only cohort which was 83.7% (similar to the entire cohort). Similarly, to demonstrate the reproducibility of our PCs in non-healthy cohorts we did the same calculation with participants who have had a previous myocardial infarction (n=671). This sub-group had 80.7% of the variation explained in the first 11 PCs. These results demonstrate that the PCs and the amount of shape variance explained are relatively robust. We report these results in the text as follows:

Results

Together, the first 11 PCs (the PCs that individually captured > 1% variance) accounted for 83.6% of the total shape variation; 83.7% in European individuals (n=41,235), 83.0% in non-Europeans (n=2,441) and 80.7% in participants with previous myocardial infarction (n=671) (Line 109).

R 2.3 - Along the same lines, the clinical value of the observations may be enhanced if specific anatomical features of the PCs (like shown in table 1) are further explored. Rather than the multifactorial (and intransparent) PCs such measures may be put into focus in further studies. A potentially interesting observation in this regard is a component of PC2, i.e. "Orientation of the valves and the length to lateral dimension ratio" which associates with "AF, AV block". Such traits and respective associations may be validated and lead to biological insights with relevance for human biology.

Response:

We agree this idea is very promising. In future work we will investigate the genetic and disease relationships of specific dimension ratios inspired by the PCs: e.g., length to lateral dimension ratio, valve orientation, lateral septal width and anterior posterior width. However, we expect some loss of power with this approach since some shape information will be potentially lost. Relationships with disease will be more informative when more disease cases are available through UK Biobank. With the current work, we establish that there are key features that explain sizeable amounts of shape variation within a mostly healthy cohort. The selection of simple clinically interpretable phenotypes are promising targets for future analyses, with their selection supported by this work. We have added the following text to the Discussion section to indicate areas for future work.

Discussion

Our study has some limitations. Firstly, the shape PCs are derived from an unsupervised machine learning method; although we manually attributed biological features to the multidimensional variation observed in the PCs, these are high dimensional traits and they cannot be reduced to a simple anatomical metric. (Line 355)

In the future, we plan to examine anatomical cardiac shape features informed by the PCs identified here and explore the direction of effect between multidimensional heart shape and cardiometabolic disease, as well as further exploring the functionality of the identified candidate genes. (Line 367)

R 2.4 - It is unclear whether the PCs were adjusted for body size. Obviously, larger people have larger hearts such that PCs may be indirectly affected by non-cardiac

traits. Indeed, PC1 (which explained most of the variability) was strongly correlated with height and inversely with age. Thus, it is unclear whether this PC is rather a function of body composition than a specific feature of the heart.

Response:

We took a similar approach as previous genetic studies of mass and volume [Aung 2019, PMID 31554410] and did not adjust the PCs for body size. Rather, the GWAS and logistic regressions of both PC score and PRS against outcome incidence were adjusted for both height and BMI, which are standard covariates used to adjust for body size. PC1 (heart size) is known to be not just a function of body size, but to also be associated with disease and risk factors and an important clinical indicator of future adverse events [Medrano-Gracia 2014, PMID: 25160814].

For completeness, we have run a series of regression analyses, regressing PC1 against a combination of height, body surface area (BSA) and BMI, to assess whether BSA is a better covariate to use in our analyses. We found similar associations between PC1 and height+BMI ($r^2=0.595$), height+BSA ($r^2=0.603$) and height+BMI+BSA ($r^2=0.609$). Therefore, we are confident that the covariates of height and BMI included in our analyses sufficiently account for the effects of body composition when reporting associations, either directly with disease or with genetics. We address this decision in the text.

Methods

We also assessed body surface area as an alternative covariate for body size: as a covariate BMI was found to perform comparably with body surface area in our phenotypes (r^2 between PC1 and height + BMI = 0.595, between PC1 and height + body surface area = 0.603. (Line 463)

R 2.5 - In this respect it would be interesting to learn whether some of 14 signals not previously reported for any cardiac trait, of which, however, 7 showed associations with non-cardiac traits, display association with measures of body size and or body composition. In fact, according to the authors only 7 loci have not been previously reported in any GWAS.

Response:

Supplementary Tables 15-17 provide details on the lookups of our reported loci in existing GWAS databases for examining relationships with other traits and diseases. Focusing on the 14 loci previously unreported for cardiac traits defined in the manuscript – the locus at *FERD3L* was previously identified for height and seated height; the locus at *C7orf25* was identified for waist to hip ratio, and the signal at *H1-0* was identified for seated height. Thus 3/14 signals demonstrate genome-wide significant associations with measures of body size or body composition. This information is provided in the paper in the Supplementary Tables, alongside all other associations we observed with both cardiac and non-cardiac traits and mentioned in the discussion (Line 295).

R 2.6 - The authors relate PCs to traditional anatomical features, i.e. descriptively, PC1 was associated with overall heart size, PC2 with apex-base length and sphericity – etc. However, it is unclear whether the (genetic) associations with the PCs also apply to these anatomical features, since the PCs explain only little of their variability.

Response:

The heritability estimations of the PCs (ranging 8.5-36.3%) demonstrated that a proportion of the variation observed can be attributed to genetic factors. However, the GWAS results indicated that the associated variants contributed a small percentage of the genetic variation (0.1-1.1%). For each PC, we observed an overlap of loci with previously reported CMR measures and cardiovascular traits, and across PCs and other traits, thus demonstrating pleiotropy. Relatively small percent of variance explained are also found in other cardiovascular phenotypes including those derived from CMR [PMIDs: 29769521, 34338756, 35697868], which is potentially due to the lower sample size of imaged participants available in the UK Biobank.

We also observe associations between the loci identified for PC1, an easily interpretable measure of heart size, and CMR-derived measures of cardiac structure and function (Supplementary Table 23). This is a clear association of the anatomical shape features with the genetic associations.

R 2.7 - Of the 45 loci with significant association to any of the PCs, only 6 were not associated with any other PC. Thus, it may be questioned as to whether the PCs reflect unique traits or have a strong overlap.

Response:

Although the PCs represent unique and distinct variations in the shape space as derived from PCA, the PCs demonstrate significant genetic correlation (Supplementary Figure 1) between each other. The genetic correlation informs us that several of the PCs have shared underlying genetics and biology, for example PCs 2 & 5. This suggests that these loci may be important in several aspects of shaping the heart and not just specific to one PC. These findings are consistent with previous observations of the genetic basis of traditional cardiac structural phenotypes (e.g. mass and volume, Pirruccello *et al* 2020 PMID: 32382064).

R 2.8 - The discussion on the potential functional effects of the affected genes is highly speculative since association with a "PC" rather than an anatomically quantifiable measure precludes any understanding what the genetic variant may do.

Response:

We believe the power of the PCs are that they are real quantifiable shape features which have meaning by capturing the largest modes of variability among hearts in the population. Any new individual's heart can be projected onto fixed eigenvectors (PC modes) to compute the PC scores for that individual.

The results from GWAS provide potential genes and mechanisms relating to the phenotype studied. Many of the PCs demonstrate significant associations with outcomes both phenotypically and genotypically, and this motivates us to try and understand the genetic underpinnings of these disease biomarkers. In the paper we describe results from a post-GWAS bioinformatic pipeline to highlight the potential candidate gene at each of the loci. However, investigating the functional mechanisms of the genetic variants and candidate genes we have discovered is beyond the scope of this paper. Future work will be required to validate the candidate genes and their functionality, and this is indicated in the Discussion section.

Discussion

In the future, we plan to examine anatomical cardiac shape features informed by the PCs identified here and explore the direction of effect between multidimensional heart shape and

cardiometabolic disease, as well as further exploring the functionality of the identified candidate genes. (Line 367)

R 2.9 - Directionality/Causality of associations are unclear. For example, some PCs of heart shape associate with diabetes mellitus. It is unclear whether the genetic/anthropometric underpinnings of the PC also affect risk of DM – or whether the subsequent manifestation of DM affects the probability to present with a specific PC.

Response:

We agree the causality of associations is not addressed and we have added this to future work. Although we excluded prior history of cardiovascular disease from the GWAS analysis, it is possible that subclinical mechanisms link shape features to subsequent manifestation of disease. The polygenic risk scoring shows the genetically predicted PCs are associated with outcomes, suggesting that genetic associations with shape features are clinically relevant and warrant further investigation. This is indicated in future work in the discussion.

Discussion

See response R2.8

R 2.10 - The authors may consider to mention that the associations observed for PCs by GWAS were explored bioinformatically but not validated experimentally.

Response:

The authors agree it is important to establish that our findings are not experimentally validated and present this as future work in our exploration of the candidate genes identified.

Discussion

See response R2.8

R 2.11 - It is also challenging to explain/understand why PC5 and its PRSs display opposing directionality with regard to risk of HCM.

Response:

We agree the difference in directionality between the PC score association and its PRS with HCM is not readily interpretable. To investigate this relationship further we attempted to replicate the PC5 PRS in a new independent HCM cohort.

We constructed our PC5 PRS in 2,284 HCM patients and used 40,825 European controls from the UK Biobank matched by age (decade) and sex. Our PRS included 7 lead SNV's from the PC5 GWAS, and two high LD proxies ($r^2 > 0.8$).

In the new HCM cohort, we found a significant positive association between the PC5 PRS and HCM: OR 1.21 (1.16-1.27, $p=1.63 \times 10^{-19}$). This result agrees with our original observation, using HCM cases from UK Biobank ($n=517$) and controls ($n=40,854$), individuals in the imaging study were excluded, where the OR was 1.29 (1.19-1.42, $p=4.20 \times 10^{-9}$).

These new results validate our findings of the association between the PC5 PRS and HCM. These new results have been added to the manuscript.

Addressing the discordance of the observational and genetic associations for PC5 and HCM, the number of HCM cases in the imaging cohort was low ($n=36$, whereas for the PRS testing

in UK Biobank the prevalence of HCM was higher (n=517 in the non-imaging European UK Biobank cohort). In this study, the discordant observational and genetically informed associations could be attributed to a severely low sample size, however observational studies are susceptible to both confounding and reverse causation, e.g., the relationship between LV mass and blood LDL levels (PMID: 33213727). We believe that the association between PC5 and HCM needs further investigation when more HCM cases are available for the imaging cohort.

Results

As there was discordance in the direction of effect between PC5 and its PRS with HCM, we sought replication of the PC5 PRS findings. We validated the positive association between PC5 PRS and HCM in an external European HCM cohort comprising 2,284 cases and UK Biobank controls, with OR of 1.21 (95% confidence intervals 1.16-1.27, $p = 1.63 \times 10^{-19}$). (Line 262)

Methods

To validate the association between the PC5 PRS and HCM, we used an external HCM cohort, comprising individuals with HCM referred to the Inherited Cardiovascular Disease unit at St. Bartholomew's Hospital, London, UK, the Inherited Cardiovascular Disease Unit at The Heart Hospital, University College London Hospitals (UCLH), London, UK and the Unidad de Cardiopatías Familiares of Complejo Hospitalario Universitario A Coruña, Spain. All patients gave written informed consent for genetic testing, and the study was approved by the regional ethics committees (London: 15/LO/0549; Coruna: 2021/182). Clinical parameters were recorded using previously described methods and were stored in a dedicated database⁸⁰. HCM was defined by the presence of a maximal left ventricular wall thickness ≥ 15 mm in probands or ≥ 13 mm in relatives. Patients with previously confirmed HCM phenocopies (e.g., Fabry disease, amyloidosis, glycogen storage diseases, and RASopathies) were excluded from the study. The samples used in this study were collected from 2011 to 2019 in London and 1991 to 2021 in Coruna. The HCM samples from the two centres (n=3,024) were genotyped at University College London Genomics unit using the Infinium Global Screening Array-24SA v3.0 with a standard methodology.

Quality-control checks on the genotype data were performed using PLINK v1.970,71. Individuals were excluded with a genotype missing rate of $> 5\%$, heterozygosity exceeding three standard deviations from the mean or inferred sex discordance. At a variant level, variants were excluded if exceeding a Hardy-Weinberg equilibrium of 1×10^{-6} or missingness $> 2\%$ (for variants with a minor allele frequency of $\leq 3\%$, a more stringent missing rate cut-off of $\geq 1\%$ was applied). Patients were checked for relatedness using $\sim 600,000$ variants using the KING relationship inference tool v2.2.6⁸¹ excluding up to and including second-degree relatives.

We constructed the PC5 PRS in 2,284 HCM patients from this dataset, and 40,825 European case controls from the UK Biobank, matched by age (decade) and sex, these individuals were not included in the UK Biobank's imaging cohort, excluding individuals carrying rare variants (MAF < 0.001) in the eight most established sarcomeric genes for HCM (MYH7, MYBPC3, TNNT2, TNNI2, ACTC1, MYL2, MYL3, TPM1) from the whole-exome sequencing data available in $\sim 250,000$ individuals at analysis. Our PRS included the lead SNV's from the PC5 GWAS, however of the 9 SNV's used in the PRS two were not present in the genetic data from the HCM cohort, therefore proxies in high LD were used instead. These were rs10155223 and rs146170154, substituted respectively with rs1500800 ($r^2=0.98$) and rs3176326 ($r^2=0.99$). We regress the standardised PRS against case-control status. (Line 651)

Reviewer #3 (Remarks to the Author):

Thank you for the opportunity to review the manuscript, "Genetic basis of right and left ventricular heart shape," by Burns and colleagues. This work aims to define and study

the principal components of variation in cardiac morphology, as defined by surface meshes derived from MRI data. This approach is novel and has the potential to offer unique insights distinct from what may be gathered from studying traditional MRI measurements. However, I think there are some areas where the manuscript requires additional details or improvement. My comments are below.

R 3.1 - Although many aspects of this paper are clearly novel, I do not think the authors can claim to be the first to study the genetic basis of heart shape. A published paper by Vukadinovic et al. (PMID: 36996817) analyzed cardiac sphericity, including GWAS. That paper should probably be cited, given the authors here spend significant text discussing cardiac sphericity as an important feature captured by their PCs. Another group (Meyer et al. PMID: 32814899) has studied ventricular fractal dimensions. Though, perhaps fractal dimensionality is distinct from shape, which is a somewhat non-specific term. In any case, I do not think the authors can claim to be the first to study the genetics of heart shape.

Response:

We have modified the text to clarify that this the first study of multidimensional shape features (see changes in R 2.1 response), which attempts to capture most of the shape variation in the population.

The authors also thank Reviewer 3 for providing additional relevant references for cardiac shape genetic relationships with disease, these have now been included.

Introduction

Previously, standard cardiac structural phenotypes have included left ventricular (LV) and right ventricular (RV) mass, volume, mass to volume ratio, sphericity **index**, conicity and myocardial strain^{4,5,6,7,8,9,10}. (Line 49)

However, these shape measures are simple one-dimensional metrics and However, they do not capture the multidimensional shape variations that can be extracted from advanced imaging examinations. (Line 58)

In this work, we discover there is a significant genetic basis of PC-derived multidimensional cardiac shape phenotypes and demonstrate associations with several prominent cardiometabolic diseases through both atlas-derived PC scores and genetically predicted shape PCs. (Line 82)

References

10, Vukadinovic, M., Kwan, A. C., Yuan, V., Salerno, M., Lee, D. C., Albert, C. M., Cheng, S., Li, D., Ouyang, D., & Clarke, S. L. (2022). Deep learning enabled analysis of cardiac sphericity. *MedRxiv*, 2022.07.20.22277861. <https://doi.org/10.1101/2022.07.20.22277861> (Line 272)

R 3.2 - One challenge of reading this paper is understanding what the PCs represent. Of course, determining the nature of PCs derived from high-dimensional data is not an easy task and requires some subjective human interpretation. However, I think the authors should provide more explanation, data, or examples to better help the reader understand each PC. The GIFs are helpful, but they are not labeled, so it is unclear which GIFs represent which PCs. Further, there should be better consistency in the manuscript. For example, Figure 2 describes PC8 simply as “LV morphology.” But

Table 1 describes PC8 as “Tricuspid opening size and LV wall thickness.” PC2 is described in the text as representing “roundness” but in Figure 2 it is labeled as “Apex-base length”, and Table 1 suggests that it represents “orientation of the valves.” The text also describes PC2, PC3, PC4, and PC5 as representing sphericity. The difficulty in knowing and describing what the PCs really represent may be a limitation of this method.

Response:

The GIF's have been labelled more descriptively as Supplementary Movies with legends.

We have harmonised descriptions across the text and figures, and included a supplementary section describing the PCs in detail.

It is also reasonable to say that the difficulty in extracting biologically meaningful features from an unsupervised learning method is a limitation of the work, and we have included this in the limitations section.

Results

Descriptively, PC1 was associated with overall heart size, PC2 with apex-base length and sphericity, PC3 with anterior-posterior width and sphericity, PC4 with relative orientation and sphericity of the RV relative to the LV, and PC5 with lateral width and sphericity. Other PCs had more complicated shape changes; for example, PC9 was associated with relative size and length of the LV relative to the RV. PCs 2, 3 and 5 also represent variations of cardiac sphericity in different dimensions (variation in length in different dimensions causes the ventricles to be more spherical). (Line 102)

Discussion (limitations)

Our study has some limitations. Firstly, the shape PCs are derived from an unsupervised machine learning method; although we manually attributed biological features to the multi-dimensional variation observed in the PCs, these are high dimensional traits and they cannot be reduced to a simple anatomical metric. (Line 355)

R 3.3 - Related to the above, since defining the PCs is central to the paper, I think it is important that all PCs are included in Figure 2. Also, why is PC11 “not assessed” in Table 1? It is also not included in Figure 3. If PC11 is not going to be analyzed whatsoever, then perhaps it should be excluded from the paper all together.

Response:

We did perform analyses of PC11 since it was above the 1% threshold of variance explained, however it provided no significant GWAS signals. We have updated Table 1, Figures 2 & 3 to include PC11 and better fit the narrative of the paper in its inclusion and eventual exclusion alongside PCs 6 & 7.

R 3.4 - The authors choose to only show PCs 1-5 and PC9 in Figure 4. They note, “In this association testing we selected only PCs with promising GWAS results for comparison with PRS associations - PCs 6, 7 and 11 had no genome-wide significant loci ($p < 5 \times 10^{-8}$) and were dropped, and PCs 8 and 10 had limited loci for PRS construction (< 3 SNV's) so were also dropped.” This section is a bit confusing because at this point in the manuscript, no GWAS results have been presented. If the genetic analyses motivate which PCs to use for outcomes analyses, then I would recommend placing the genetic analyses first.

The manuscript has been changed to include illustration of all PCs in Figure 4. We now include PCs 6, 7 and 11 in analyses up to and including the GWAS, and those GWAS results inform our exclusion of PCs 6, 7 and 11 from subsequent analyses.

Results

~~In this association testing we selected only PCs with promising GWAS results for comparison with PRS associations — PCs 6, 7 and 11 had no genome-wide significant loci ($p < 5 \times 10^{-8}$) and were dropped, and PCs 8 and 10 had limited loci for PRS construction (< 3 SNV's) so were also dropped.~~

R 3.5 - The methods section for the logistic regression testing the association between PCs and outcomes is not detailed. What covariates were included? Do these results represent a multivariable model with all PCs included (ie are the odds ratios the independent effects of each PC)? Or this simply a univariable analysis of each PC versus each outcome? If the results in Figure 4 only show univariable analyses, then it would be useful to also describe in text or show in a figure which PCs independently predict which diseases.

Response:

These results represented a series of multivariable models including only one PC and a fixed set of covariates. In considering this comment we have changed the method of covariate selection used in the analysis. We now use a stepwise regression method to select covariates significant to each cardiometabolic outcome selected for the study. We then use these covariates in our linear regressions on each outcome. In ST3-ST9 we present our stepwise regressions and associations between PCs and outcomes for both PC scores and PRS, for both the imaging cohort and full UK Biobank cohort. We make this distinction because the PC scores come from participant data taken at imaging, however the genetic data comes from the baseline visit. In these tables we present both univariate and multivariate (including significant covariates) associations between PC score/PRS and outcome.

All relevant tables and figures have been updated to include this method of covariate selection.

Methods

This regression was adjusted for covariates selected using stepwise regression to identify covariates significant for each outcome, selecting from: age, sex, height, BMI, alcohol consumption, adjusted systolic and diastolic blood pressures, smoking status, low density lipoprotein and mean triglyceride levels (Supplementary Tables 3-9 (A)). We also assessed body surface area as an alternative covariate for body size: as a covariate BMI was found to perform comparably with body surface area in our phenotypes (r^2 between PC1 and height + BMI = 0.595, between PC1 and height + body surface area = 0.603. For testing associations between PCs and outcomes, the imaging visit data from the UK Biobank was used. Missing values were imputed for these covariates using MICE⁶⁹ in R, each imputed covariate had < 5% missing data. High density lipoprotein level was considered as a covariate and dropped as it had > 5% missing data. (Line 459)

covariates selected to be significant for each outcome through stepwise regression (see method in Correlation and association analysis, Supplementary Table 3-9 (B)), alongside genetic array used and the first 15 genetic principal components. For these regressions the first visit UK Biobank data was used. (Line 629)

R 3.6 - OR plots should use a log scale for the X-axis and be centred on 1. Otherwise,

protective effects appear weaker than risk effects despite having the same magnitude of beta.

Response:

We have modified the figure as suggested.

R 3.7 - Subjects with prevalent MACE were excluded. How was MACE defined and derived?

Response:

MACE was derived from the UK Biobank's hospital episode statistics (HES data) using ICD9 and ICD10 codes, alongside self-reported incidences of cardiovascular events. This has been added into the supplementary methods and referenced in the text.

Supplementary methods: Major adverse cardiovascular event (MACE) definition

The study cohort used in the genome-wide association study selected healthy European UK Biobank participants with no recorded major adverse cardiovascular events. This was defined using hospital episode statistics data ICD10 codes: I20.0, I21, I22, I23, I24, I26, I30, I40, I41, I33, I38, I43, I46, I51, I44, I45, I47, I48, I49, I50, I60, I61, I62, I63, I64, I69, I710, I711, I713, I715, I718, I42, I13, I14. We also use ICD9 codes 4109, 4280, 4281, 4289. Finally, we also exclude participants with self-reported previous major adverse cardiovascular events from the baseline questionnaire, as well as participants with either self-reported or algorithmically defined myocardial infarction or heart failure using the relevant UK Biobank data fields. (Line 63)

Methods: Genetics cohort quality control

To avoid the confounding effects of cardiovascular disease phenotypes the cohort was restricted to a model calculated using LV ejection fraction $\geq 40\%$ ($n=1695$ excluded) and no previously reported major adverse cardiovascular events (defined in Supplementary Methods) (Line 440)

R 3.8 - The authors described GTCA analysis, writing, "Conditional analyses with genome-wide complex trait analysis (Methods) indicated two independent signals..." I believe they mean to say, "two additional independent signals."

Response:

The authors thank the reviewer for this correction, we have updated the text. (Line 151)

Reviewers' comments:

Reviewer #2 (Remarks to the Author):

The authors made an attempt to address my concerns, however my main concern remains.

Genetic associations with "principle components", i.e. measures that are neither transparent in how exactly they were built, nor measurable by any clinical test (including MRI) are almost non-interpretable from a clinical point of view.

Many details on how to interpret the findings remain unclear. I try illustrate this by some answers given by the authors in parenthesis below.

The authors agree "that the identified PCs are dependent on the dataset" they used, i.e. it seems that the data are not reproducible at the present time.

The authors agree "the causality of associations is not addressed", i.e. a main advantage of "traditional" genetic association studies was questioned by the authors.

The author agree "that the difference in directionality between the PC score association and its PRS with HCM is not readily interpretable."

Reviewer #3 (Remarks to the Author):

In this revision, the authors adequately addressed several of my prior concerns. However, I do not feel my previous comments 1 and 2 were addressed. My concerns are detailed below.

In response to my previous comment 1, the authors modified their claim about being the first to study the genetics of cardiac shape to now state they are the first to study the genetics of PC-derived shape. However, I found the rest of their response quite peculiar. In my comment, I provided PMIDs for two published papers that have already demonstrated the value of genetic studies of cardiac shape using MRI. The authors chose not to cite one of these manuscripts (Meyer et al. PMID: 32814899) without explanation. I will assume that they feel strongly that this work is not relevant to theirs, despite it being a clear example of the same concept (genetic analysis of nonstandard cardiac MRI phenotypes) applied to the same population/data. However, the second paper Vukadinovic et al. PMID: 36996817) is undoubtedly relevant as it is the first to study the genetics of cardiac sphericity and it shows the association with AF and cardiomyopathy.. Rather than cite this manuscript, the authors chose to reference a preprint. Seeing this choice made me weary of the authors' intentions, and I therefore scrutinized their references more closely. I now get the impression that they are hoping to sweep the prior cardiac shape studies under the rug in order to pass off their findings as more novel than they are. In the Abstract, Introduction, Results, and Discussion, the authors harp on the importance of cardiac sphericity as a phenotype, and they highlight the association with AF as a main result. The concept of cardiac sphericity dates back to at least the 1990s (eg PMID: 9396419), and the association with AF was first observed by Ambale-Venkatesh et al. (PMID: 27694110) and then demonstrated in the much larger UK Biobank cohort by Vukadinovic et al. In the Introduction, the authors include sphericity as a "standard" structural phenotype (though clinically it is not considered a standard measurement), but when they list the known associations with standard phenotypes, they do not mention sphericity and its associations with AF and cardiomyopathy. In the Discussion, they

discuss some of the prior sphericity papers but fail to mention the Vukadinovic paper, which is the one most relevant to their analyses, as it studies the exact same population using the same MRI data.

In response to my previous comment 2, the authors provided labeled movies and improved the consistency of the descriptions of each PC. However, the core critique remains. The authors are not able to describe what each PC represents. For example, PC8 is simply described as "LV morphology." It appears some PCs are somewhat readily interpretable while others are not. Inability to interpret PCs is an inherent limitation of this approach. I suspect the authors do not know what some of the PCs represent. I certainly do not know after watching the videos. Thus, they essentially report genetic associations for unknown phenotypes. If the authors are uncertain about any of the PCs, they should be transparent about this fact. For example, in the second paragraph of the Results, simply note which PCs do not have a clear interpretation. If the authors do feel they have a clear understanding of each PC, it is essential to provide these explanations. Without understanding the phenotype, the downstream analyses provide little insight.

RESPONSE TO THE REVIEWERS

We wish to thank the reviewers for their further comments on our manuscript, we provide a point-by-point response below:

Reviewer 2

1. "Genetic associations with "principal components" (are) measures that are neither transparent in how exactly they were built, nor measurable by any clinical test (including MRI) are almost non-interpretable from a clinical point of view."

We have shown that principal components are indeed transparent in how they are built, are quantifiable by MRI and other imaging methods, and are interpretable with respect to clinical outcomes. Firstly, the PCs are transparent in how they are built, since they arise from a well understood pipeline using automated image analysis, model customization and statistical atlasing as shown in Figure 1. Each step is explainable and transparent. Secondly, the PCs are readily quantifiable by MRI and other imaging methods, simply by following the pipeline to segment the images and customize the shape model, and then by projecting this shape model onto the 11 PC modes. This can be performed automatically at imaging, so the PC scores would be available as part of the imaging report. Thirdly the PCs are interpretable with respect to clinical outcomes since they are both directly associated with cardiometabolic diseases, and the polygenic risk scores associated with PCs are also associated with cardiometabolic diseases. This implies that the genetic factors underlying shape variation in the population are also involved in disease processes. This has been clarified in the Discussion.

2. "The authors agree "that the identified PCs are dependent on the dataset" they used, i.e. it seems that the data are not reproducible at the present time."

While we agree that any principal component analysis is by definition dependent on the dataset, we went on to say that "the large cohort size (~40k) and limited number of PCs (11) studied in this work gives us confidence that these PCs are robustly characterized and would be generalizable to other cohorts". In other words, these PCs would also capture a substantial amount of shape variation in other cohorts. In particular, we addressed this issue by performing additional experiments that demonstrate that the PCs and the amount of shape variance explained are relatively robust: "Comparison with another cohort is more typically performed by projecting the new heart shapes onto the existing PC modes to derive the new shape scores with respect to the existing atlas, rather than deriving new PC modes. In this approach the question becomes how well the existing PC capture shape variation in the new cohort. As a surrogate test of reproducibility we computed the % variance explained in heart shape by our PC's in non-European and MI sub-cohorts, to demonstrate the degree to which the 11 PCs capture heart shape. The first 11 PC's captured 83.6% of the variation over the entire cohort. Considering just the non-European individuals (N=2441), the variance explained by the first 11 PC's was 83.0%, only slightly less than the European-only cohort which was 83.7% (similar to the entire cohort). Similarly, to demonstrate the reproducibility of our PC's in non-healthy cohorts we did the same calculation with participants who have had

a previous myocardial infarction (N = 671). This sub-group had 80.7% of the variation explained in the first 11 PC's. These results demonstrate that the PCs and the amount of shape variance explained are relatively robust."

3. "The authors agree "the causality of associations is not addressed", i.e. a main advantage of "traditional" genetic association studies was questioned by the authors."

We believe the causal mechanisms of associations required a substantial amount of future work, and was not the focus of the present work, which aims at investigating the genetic architecture of heart shape, estimated using PCs. Our response was "The polygenic risk scoring shows the genetically predicted PC's are associated with outcomes, suggesting that genetic associations with shape features are clinically relevant and warrant further investigation. This is indicated in future work in the discussion." We note that several previous papers (traditional GWAS studies) have established genetic links with cardiac structural phenotypes and brain in which the causal mechanism is unknown, for example, PMIDs: 30305740 and 32814899.

4. "The authors agree "that the difference in directionality between the PC score association and its PRS with HCM is not readily interpretable."

To address this result, we sought access to an external cohort HCM to test the robustness of the association observed in UK Biobank. We showed that "These new results validate our findings of the association between the PC5 PRS and HCM". The results quoted were extensive and detailed, and can be read in the previous response.

Reviewer 3

1. "In response to my previous comment 1, the authors modified their claim about being the first to study the genetics of cardiac shape to now state they are the first to study the genetics of PC-derived shape. However, I found the rest of their response quite peculiar. In my comment, I provided PMIDs for two published papers that have already demonstrated the value of genetic studies of cardiac shape using MRI. The authors chose not to cite one of these manuscripts (Meyer et al. PMID: 32814899) without explanation".

This paper deals with trabeculation, which is not captured by the current analysis and has no overlap in terms of interpretation. We omitted this explanation from the response to reviewers, which was an oversight. We have included it in the new revised manuscript.

2. "However, the second paper (Vukadinovic et al. PMID: 36996817) is undoubtedly relevant as it is the first to study the genetics of cardiac sphericity and it shows the association with AF and cardiomyopathy. Rather than cite this manuscript, the authors chose to reference a preprint."

We apologise for the inaccurate citation to Vukadinovic et al, which was an oversight. This is corrected in the revision.

3. "The concept of cardiac sphericity dates back to at least the 1990s (eg PMID: 9396419), and the association with AF was first observed by Ambale-Venkatesh et al. (PMID: 27694110) and then demonstrated in the much larger UK Biobank cohort by Vukadinovic et al. In the Introduction, the authors include sphericity as a "standard" structural phenotype (though clinically it is not considered a standard measurement), but when they list the known associations with standard phenotypes, they do not mention sphericity and its associations with AF and cardiomyopathy. In the Discussion, they discuss some of the prior sphericity papers but fail to mention the Vukadinovic paper, which is the one most relevant to their analyses, as it studies the exact same population using the same MRI data".

We cited several papers on sphericity (references 60-63) which included Ambale-Venkatesh et al. (PMID: 27694110) and the association with AF. We believe the paper by Wong et al PMID: 15541243, which we cited, is a more direct illustration of the relationship post MI than PMID: 9396419 which the reviewer indicates, although we have now included this as well in the revision (Introduction), along with citing Vukadinovic in the Discussion. Although sphericity changes are readily observable in some of the PCs, they are not exactly the same as sphericity, as evidenced by the different genetic signals obtained in the current studies over those of Vukadinovic et al. This has been clarified in the Discussion.

4. "I now get the impression that they are hoping to sweep the prior cardiac shape studies under the rug in order to pass off their findings as more novel than they are."

We think this comment is highly unfair and hope we have demonstrated that this is definitely not the case. Although previous cardiac shape studies have captured useful and

substantial amounts of shape variation, the PCs proposed and developed in our work capture a substantial amount of shape variation that was not reflected on the previous shape phenotypes. This has been clarified in the Discussion.

5. “The authors are not able to describe what each PC represents... If the authors are uncertain about any of the PCs, they should be transparent about this fact”.

We were very transparent about this in the previous revision (“Since PCA is an unsupervised method of dimensionality reduction, the PCs are not readily interpreted cardiac morphological features”). However, we also made clear that many of the PCs have intuitive descriptions as detailed in the results section. The PCs are useful phenotypes because they explain a substantial proportion of the shape variation, and we demonstrate that the PCs are associated with cardiac remodelling and outcomes. Also the polygenic risk scores derived from the PCs are associated with cardiometabolic disease. Although further work is required to fully understand the genetic basis of the PCs, this is common for genetic evaluations for many inherited traits. In our opinion, our work investigating the genetic background of the PCs has informed on new genomic regions and pathways that will enable further research into the interpretation of the PCs and cardiac shape.

REVIEWER COMMENTS

Reviewer #2 (Remarks to the Author):

We went through several rounds of reviews with this paper. The principle content has not changed.

My main criticisms likewise remain. From a clinical as well as a conceptual point of view, a genetic association with Principal Components - that cannot be clearly explained in what they represent (except a mixed bag of diverse measurements) - may be confusing and useless.

It is likely that well established clinical measurements fully capture the information provided by the PCs - but this cannot be tested.

The authors continue to admit that the causality between the genetic association with Principal Components cannot be tested. I.e., there is a chance of reverse causality.

The authors continue to admit that the data are not reproducible, given that no other data sets with respective source data exist.

Reviewer #4 (Remarks to the Author):

The authors present a GWAS analysis on shape principal components obtained from biventricular meshes, derived in turn from cardiovascular magnetic resonance (CMR) imaging through an automatic segmentation and registration pipeline.

Methodological novelty: the idea of using unsupervised phenotypes derived from 3D meshes in GWAS, and in particular shape PCs, is not novel, and it has been explored in the context of cardiac imaging genetics by a recently published paper (PMID=38523678), and previous conference communication (see ref. below). The authors may also consider citing another paper which also uses PCA for facial shape in the context of genetic association studies (PMID=29459680).

In spite of this lack of methodological novelty with respect to said work, the authors implement PCA on a biventricular mesh as opposed to simply a left-ventricular mesh, which seems to lead to additional discoveries, however to justify its publication the following additional analyses would be required.

Previously found genes: the authors may consider updating the novelty claims with respect to the found loci to take into account those that have already been found by the previously cited paper (PMID=38523678), for instance the CCDC91, GMDS and EN1 loci. Some of the associations in Table 1 are absent from the previously cited paper (i.e. they are indeed novel), probably because the latter focused on LV. Have the authors considered performing shape PCA on each chamber separately and compare the association results? This analysis would help establish the implemented biventricular PC analysis as the cause for the additional discoveries (unless it is rather the different segmentation and registration pipeline), and also help with further localizing the genetic associations to one ventricle or another. To this effect, a comparison of the SNP effect size estimates can be carried out.

GWAS on simpler parameters: the authors mention previous studies on simpler parameters such as volumes, myocardial mass and quantities derived therefrom. It would be interesting to see which of the found loci are discovered for each of these parameters, using the same meshes to retrieve them to make sure that the segmentation and registration methods, as well as the sample size, do not

influence the outcome. This additional analysis would underscore the advantage of using shape PCs in this context. The parameters for which each of the loci are found with statistical significance could then be added as an extra column to Table 1.

p-value significance threshold: To control for false discovery rate, further adjusting the p-value significance threshold by dividing by the number of tested PCs, or identifying associations with $5e-8 / 11 < p < 5e-8$ as suggestive, may be more conservative.

PC-disease associations: have you considered using Mendelian Randomization to provide evidence of the potential causal role of the shape PCs on the different disease outcomes being studied?

Other comments:

Line 235: where it reads CCD91, should it read CCDC91?

Line 350: The authors mentioned that the PLN locus is not discovered in the PC-based analysis. However, based on the position of the locus annotated as SLC35F1, PLN could be a potential causal gene for this association. The neighboring PLN gene can be seen in the LocusZoom plot for this locus.

Line 591: Is this the number of genes or the number of gene-tissue pairs?

Figure 2 (effect of PC on shapes): what is the meaning of the sign? Does the color represent some sort of signed radial distance from the centroid?

References:

R. Bonazzola, E. Ferrante, N. Ravikumar, Y. Xia, B. Keavney, S. Plein, T. Syeda-Mahmood, A.F. Frangi. Unsupervised ensemble-based phenotyping enhances discoverability of genes related to left-ventricular morphology. *Nature Machine Intelligence* (2024). PMID=38523678.

P. Claes, J. Roosenboom, J. D White, T. Swigut, D. Sero, J. Li, et al. Genome-wide mapping of global-to-local genetic effects on human facial shape. *Nature Genetics* (2018). PMID=29459680.

R. Bonazzola, N. Ravikumar, R. Attar, E. Ferrante, T. Syeda-Mahmood, A. F Frangi. Image-derived phenotype extraction for genetic discovery via unsupervised deep learning in CMR images. *International Conference on Medical Image Computing and Computer-Assisted Intervention (MICCAI)*, 699-708 (2021).

Reviewer #5 (Remarks to the Author):

Genetic basis of right and left ventricular heart shape - Response to Reviewers

The authors would like to thank the Reviewers for their rigorous and insightful review of the research. We have addressed the comments in order, and highlight the relevant changes made in the manuscript. The comments are numbered by both Reviewer and comment, (e.g., R4.1 for Reviewer 4, point 1, etc) and labelled in highlighted changes in the manuscript. In this document, we indicate the page and line numbers where we have made insertions into the text, and where we have removed text.

Reviewer #2 (Remarks to the Author):

We went through several rounds of reviews with this paper. The principle content has not changed.

My main criticisms likewise remain. From a clinical as well as a conceptional point of view, a genetic association with Principal Components - that cannot be clearly explained in what they represent (except a mixed bag of diverse measurements) - may be confusing and useless.

It is likely that well established clinical measurements fully capture the information provided by the PCs - but this cannot be tested.

The authors continue to admit that the causality between the genetic association with Principal Components cannot be tested. I.e., there is a chance of reverse causality.

The authors continue to admit that the data are not reproducible, given that no other data sets with respective source data exist.

Response: We thank the Reviewer for their time reviewing our manuscript, and we continue to disagree with their appraisal of our work. Briefly, the paper shows that i) genetic associations with PC scores enable investigation of novel mechanisms that underlie large variations of heart shape in the population; ii) this information is not captured by established clinical measurements or other standard MRI metrics; iii)

causality can be tested (and results are shown in this revision); iv) reproducibility can be tested by projecting new cohorts onto the modes generated with the existing cohort (as imaging data become available).

Reviewer #4 (Remarks to the Author):

R4.1 *The authors present a GWAS analysis on shape principal components obtained from biventricular meshes, derived in turn from cardiovascular magnetic resonance (CMR) imaging through an automatic segmentation and registration pipeline.*

Methodological novelty: the idea of using unsupervised phenotypes derived from 3D meshes in GWAS, and in particular shape PCs, is not novel, and it has been explored in the context of cardiac imaging genetics by a recently published paper (PMID=38523678), and previous conference communication (see ref. below). The authors may also consider citing another paper which also uses PCA for facial shape in the context of genetic association studies (PMID=29459680).

In spite of this lack of methodological novelty with respect to said work, the authors implement PCA on a biventricular mesh as opposed to simply a left-ventricular mesh, which seems to lead to additional discoveries, however to justify its publication the following additional analyses would be required.

Previously found genes: the authors may consider updating the novelty claims with respect to the found loci to take into account those that have already been found by the previously cited paper (PMID=38523678), for instance the CCDC91, GMDS and EN1 loci. Some of the associations in Table 1 are absent from the previously cited paper (i.e. they are indeed novel), probably because the latter focused on LV.

R4.1 response: The authors thank the Reviewer for such a detailed and thoughtful response and review of the work. We submitted our work to Nature Communications in November 2022, so our paper was most likely in the review process at the same time as Bonazzola *et al.* The conference paper (MICCAI 2021) reported auto-encoder phenotypes, although they mention they found no significant GWAS associations using PCA. We therefore we compare our results with the summary statistics in Bonazzola *et al* 2024 in the Discussion, and highlight differences in methodology as detailed below.

Regarding the paper on face shape GWAS (Cleas *et al* 2018), we thank the Reviewer for bringing this to our attention. We have cited the paper as an example of a similar data-driven approach in a different application. Cleas *et al.* apply a hierarchical spectral clustering of facial shape resulting in a segmental decomposition, followed by PCA on each segment. This methodology differs from the current paper since the face enables a rich set of landmarks that can be locally clustered, unlike the heart which requires a more global approach.

Changes made to the paper: Page 14, lines 393-395

Bonazzola *et al.* explored the genetic basis of left-ventricular shape at end-diastole (both through PCA and autoencoder latent variables). Their PCA analysis of LV shape has similarities to our work, but there are significant differences between Bonazzola *et al.*'s methodology and ours:

1. Their LV models contain no information from the long-axis CMR images captured in the UK Biobank; instead, the LV is fully informed by the short-axis view. In our analysis, we capture information from the four-chamber, two-chamber, and three-chamber long-axis cine images. This enables the identification of the precise location of the mitral, tricuspid, and aortic valves, which are obtained from the deep learning automated image analysis.
2. Our biventricular shape models contain information on both left and right ventricles, providing information on the inter-ventricular septum (precise location and extent) and relative orientation of the ventricles in 3D space.

These methodological differences likely explain the differences in genetic variants identified across the two manuscripts. Noting the differences in the methodology, we have performed a formal look-up of our loci in the Bonazzola *et al* GWAS results to determine the overlap. The look-ups were done in Table 1 for UPE and in available summary statistics for the shape and reconstructed LV phenotypes. In our manuscript, we report 43 loci (45 independent genetic signals). A look-up of our lead variants and close proxies ($r^2 > 0.7$) in either UPE, LV shape or reconstructed LV CMR GWAS reported by Bonazzola *et al* indicated 19 signals are shared at genome-wide significance ($P < 5E-08$). Additionally, one signal (PC9: *NKX2-5*) demonstrated a suggestive association ($P < 1E-06$) with a Bonazzola locus (*NKX2-5*), this variant was in low LD ($r^2 = 0.18$) with the Bonazzola lead variant. For the remaining 25 signals our lead variants were not in LD ($r^2 < 0.1$) with any locus reported in Bonazzola *et al*. We observed support ($P < 0.05$) for 22 of our loci in the Bonazzola dataset. There was no support for 3 loci (PC1- *BAG3*, PC3 – *PRDM6*, and PC5- *SIM2*).

In view of the differences between the atlases, the results from the comparison of genetic results and timing of the papers being under review at the same time, we now

include new text in the discussion describing the publication of the Bonazzola *et al* paper and comment on the overlapping loci.

Changes made to the paper

1. Table 1: We have added an ¥ to loci that were also reported by Bonazzola *et al.* whilst our manuscript was under review (noting results were presented in Table I UPE and summary statistics).
2. In the Discussion, we describe the work of Bonazzola *et al.*, which was published while our paper was under review, and describe the differences in methodology, and overlapping loci. Discussion: Page 14, lines 379 – 393.

R4.2: *Have the authors considered performing shape PCA on each chamber separately and compare the association results? This analysis would help establish the implemented biventricular PC analysis as the cause for the additional discoveries (unless it is rather the different segmentation and registration pipeline), and also help with further localizing the genetic associations to one ventricle or another. To this effect, a comparison of the SNP effect size estimates can be carried out.*

R4.2 response: Thank you for this suggestion. We were reluctant to do this as we expected we would get different PC components with separate chamber PCAs since the resulting shape modes depend on the relative variation of the particular points used in the PCA. To test this, we generated an LV atlas at ED using our modelling pipeline (by extracting those 3D points on the biventricular shape model that are associated with the LV (i.e. LV free wall epi- and endocardial points, and septal LV and RV endocardial points). We performed genetic analyses on the resulting LV PCs (which were indeed different from the bi-ventricular PCs). We discovered 38 loci across all LV PCs. Next, we compared loci from our new LV shape atlas with the genetic results by Bonazzola *et al* (both from autoencoder latent features and PC shape atlas/reconstructed LV phenotypes). A look-up of our lead variants and proxies (using $r^2 > 0.8$) indicated 22 shared loci at genome-wide significance ($P < 5E-08$) across all phenotypes. There were 14 loci shared between our LV shape atlas and the Bonazzola *et al* PC shape atlas, with 2 of these loci also genome-wide significant for the Left Ventricular Mass Volume Ratio phenotype. A review of the 16 loci not shared indicated that 14 loci were novel to our LV PC atlas, and two were shared with our ED bi-ventricular PC atlas. These findings were

in line with expectations due to differences in the PC modes between atlases. We therefore decided not to include them in the revision since the methodological differences between the bi-ventricular atlas PCs and our LV PCs, which are also different from Bonazzola *et al*, would require extensive explanation.

R4.3 *GWAS on simpler parameters: the authors mention previous studies on simpler parameters such as volumes, myocardial mass and quantities derived therefrom. It would be interesting to see which of the found loci are discovered for each of these parameters, using the same meshes to retrieve them to make sure that the segmentation and registration methods, as well as the sample size, do not influence the outcome. This additional analysis would underscore the advantage of using shape PCs in this context. The parameters for which each of the loci are found with statistical significance could then be added as an extra column to Table 1.*

R4.3 response: We thank the Reviewer for the suggestion. However, we do not think this analysis will add new information to our paper since analyses of volumes and mass have been published in several papers (PMIDs: 31554410, 32382064, 36944631, 35697867). In our submission, we reported look-ups of our loci in published GWAS from cardiac traits available in the GWAS catalogue and other available summary statistics datasets. From this work, we demonstrated some overlap of our loci with structural and functional CMR measures (*Supplementary Tables S21 and S22*), as expected. The work of Bonazzola *et al* observed similar findings. We believe that the difference in using PC components compared to volume and mass is apparent in both our work and in Bonazzola *et al*.

R4.4 *p-value significance threshold: To control for false discovery rate, further adjusting the p-value significance threshold by dividing by the number of tested PCs, or identifying associations with $5e-8 / 11 < p < 5e-8$ as suggestive, may be more conservative.*

R4.4 response: The shape PCs are considered distinct phenotypes in our study; they are uncorrelated with each other (due to the nature of PCA) and, therefore, are expected to capture distinct biological variations. For this reason, we do not assimilate our results into one single set of results presenting variants that affect the biventricular shape. Instead, we adopt a more granular approach of investigating specifically what

each component represents alongside its biological context. Hence, we did not adjust the p-value significance threshold for our GWAS.

R4.5 PC-disease associations: have you considered using Mendelian Randomization to provide evidence of the potential causal role of the shape PCs on the different disease outcomes being studied?

R4.5 response: We performed two-sample Mendelian Randomization (MR) to investigate the relationships between cardiac shape and outcomes, focusing on the most significant cardiac shape–disease relationships observed in the manuscript (PC1 and Type 2 diabetes; PC3 and PC5 with atrial fibrillation). We found support for a negative causal association between type 2 diabetes (exposure) and PC1(outcome) (IVW: OR 0.595, 95% CI 0.228-0.962, P-value = 0.0074) which was robust in sensitivity analyses.

We found no significant support for a causal association between AF and either PC's 3 or 5, and also no support for a relationship of PC1 (exposure) affecting type 2 diabetes (outcome) incidence. Full values from these analyses have been added to *Supplementary Table 27*.

Changes made to the paper

Results: page 11, line numbers 286 – 296.

Discussion: page 13, line numbers 348-351

Methods: page 26, line numbers 737-755

R4.6 Other comments:

Line 235: where it reads CCD91, should it read CCDC91?

R4.6 response: Thank you, we have made this change in the manuscript.

R4.7 Line 350: *The authors mentioned that the PLN locus is not discovered in the PC-based analysis. However, based on the position of the locus annotated as SLC35F1, PLN could be a potential causal gene for this association. The neighboring PLN gene can be seen in the LocusZoom plot for this locus.*

R4.7 response: We have amended the discussion text to better reflect the point being made; the reviewer is correct that *PLN* was at the locus of the identified variant at

SLC35F1, and the Vukadinovic GWAS does identify a variant annotated to *PLN* which was in LD ($r^2 = 0.4$) with our signal. It was an error to say that this variant was not found in their work, this has now been removed. Two of their 4 sphericity variants are found in our GWAS, and there are 18 other sphericity-associated variants from our PC's which they do not report.

Discussion: page 13, line numbers 359-368

R4.8 *Line 591: Is this the number of genes or the number of gene-tissue pairs?*

R4.8 response: It is gene-tissue pairs, this has been amended in the text.

Methods: page 22, line numbers 621-622

R4.9 *Figure 2 (effect of PC on shapes): what is the meaning of the sign? Does the color represent some sort of signed radial distance from the centroid?*

R4.9 response: Yes the colour indicates a distance from the mean shape in either a positive or negative direction relative to the plotted Eigenvector. For PC1 for example the red indicates areas that are larger (further in the positive direction of the Eigenvector). This is better reflected in the Supplementary Movies.

Response to Reviewers

Reviewer 4

The authors thank the reviewers for their favourable comments regarding the work, and we are happy to address the outstanding comments provided.

In lines 380-381, you claim that “[Bonazzola et al.] only used short-axis information to guide mesh fitting”. However, upon examination of the corresponding segmentation algorithm (PMID: 35665663) which the authors of said paper claim to have used, it seems that long-axis views are indeed used to guide the mesh fitting. Could the authors either provide a source for this statement or modify it accordingly?

As stated in Xia et al (PMID: 35665663) “We would like to highlight that the cardiac atlas mesh used in the study (available from Rodero et al. (2021)) includes the base of the aorta and pulmonary artery, and all cardiac valve planes in addition to all four cardiac chambers. However, as the manual delineations available for the UK Biobank cohort do not include these structures, the deformation of these additional structures is driven purely by the TPS-based mesh warping step, when generating the subject-specific meshes. Thus, the aortic vessel surface visible in Fig. 2 for example, is inferred based on the predicted deformations for the adjacent cardiac chambers alone.”

Our close reading of that paper indicates that left and right atrial contours from the four-chamber and two-chamber views were included, but not ventricular contours from any long axis view, or any contours from the three-chamber view, which is the only view delineating the aortic valve. Although these views are included in the neural network which predicts geometry from the images, the mesh customization process used for training the network did not utilize ventricular contours or valve information from the long axis views. We have summarised this by adjusting the text to read “These models did not utilize valve locations from long-axis CMR images, and did not include long axis ventricular contours to guide mesh fitting, contrary to the current work in which long axis ventricular contours and valve landmarks (in particular including the aortic valve) were included from the automated image analysis of two- three- and four-chamber long axis cine images.”

Lines 390-392: I believe the mention of the Claes et al. article (GWAS on face shape features) should be placed somewhere else. Up to this point, this paragraph is committed to comparing the associations found by the authors with those from the Bonazzola et al. paper. However, the subsequent mention of the Claes et al. paper alters the flow of this discussion. How about moving this sentence to the Introduction?

Thank you for the suggestion. We have now moved these the reference to Claes et al study to the third paragraph of the introduction, and can be found on page 3 and lines 73-75.